# Tracer test modeling for characterizing heterogeneity and local scale residence time distribution in an artificial recharge site

Cristina Valhondo[1,2,3], Lurdes Martínez-Landa[2,3], Jesús Carrera[1,3], Juan J. Hidalgo[1,3], Isabel Tubau[1,3], Katrien De Pourcq[1,3], Alba Grau-Martínez[4,5], and Carlos Ayora[1,3]

[1]Institute of Environmental Assessment and Water Research (IDAEA), CSIC, C/Jordi Girona 18, 08034 Barcelona, Spain.
[2]Department of Civil and Environmental Engineering, Universitat Politècnica de Catalunya (UPC), Jordi Girona 1-3, 08034 Barcelona, Spain.
[3]Associated Unit: Hydrogeology Group (UPC-CSIC)
[4]Grup de Mineralogia Aplicada i Geoquímica de Fluids, Departament de Cristal.lografia, Mineralogia i Dipòsits Minerals, SIMGEO UB-CSIC, Facultad de Geologia, Universitat de Barcelona (UB), C/ Martí i Franquès, s/n - 08028 Barcelona, Spain.
[5]Comunitat d'usuaris d'Aigües del delta del Llobregat. Av. de la Verge de Montserrat 133, 08820 El Prat del Llobregat, Barcelona, Spain.

*Correspondence to:* Cristina Valhondo (cvaqam@idaea.csic.es)

**Abstract.**

Artificial recharge of aquifers is a technique for improving water quality and increasing groundwater resources. Understanding the fate of a potential contaminant requires knowledge of the residence times distribution (RTD) of the recharged water in the aquifer beneath. A simple way to obtain the RTDs is to perform a tracer test. We performed a pulse injection tracer test in an artificial recharge system through an infiltration basin to obtain the breakthrough curves, which yield directly the RTDs. The RTDs turned out to be very broad and we used a numerical model to interpret them, to characterize heterogeneity, and to extend the model to other flow conditions. The model comprised nine layers at the site scaled to emulate the layering of aquifer deposits. Two types of hypotheses were considered: homogeneous (all flow and transport parameters identical for every layer) and heterogeneous (diverse parameters for each layer). The parameters were calibrated against the head and concentration data in both model types, which were validated quite satisfactory against 1,1,2-Trichloroethane and electrical conductivity data collected over a long period of time with highly varying flow conditions. We found that the broad RTDs can be attributed to the complex flow structure generated under the basin due to threedimensionality and time fluctuations (the homogeneous model produced broad RTDs) and the heterogeneity of the media (the heterogeneous model yielded much better fits). We conclude that heterogeneity must be acknowledged to properly assess mixing and broad RTDs, which are required to explain the water quality improvement of artificial recharge basins.

## 1 Introduction

The need to satisfy the increasing demand for water is the main driver behind managed aquifer recharge, which is becoming a standard technique for replenishing and/or enhancing groundwater resources. One of the goals of managed aquifer recharge

is to provide aquifers with good water quality, even when lesser quality water is used to recharge the aquifer (e.g., treatment plant effluents or runoff water).

Water quality is enhanced during passage through soil (Bouwer, 2002; Greskowiak et al., 2005) because the passage causes reduction not only in turbidity and suspended matter, but also in the concentrations of dissolved organic matter (Vanderzalm
et al., 2006), nutrients (Bekele et al., 2011), pathogens (Dillon et al., 2006), and some trace organic contaminants (Dillon et al., 2006; Hoppe-Jones et al., 2010; Valhondo et al., 2014, 2015). The appropriate management of artificial recharge systems requires an understanding of the fate of the potential contaminants. This is especially relevant for recharge through infiltration basins or river bank filtration, which typically involve larger volumes of poorer quality water than typically used for injection wells.

The fate of contaminants depends on hydraulics, which is the focus of this work, and biochemistry. Hydraulics control the residence time distribution (RTD) of the recharged water reaching pumping wells, which is required to (1) ascertain the removal of potential contaminants, (2) interpret removal observations to obtain parameters describing field reaction rates and transport, and (3) foresee (and eventually correct) future changes in groundwater quality.

Understanding hydraulics entails an understanding of the spatial and temporal distributions of water fluxes around the
recharge system and the relationship between the recharge system and the aquifer (i.e., recharge affected area, mixing of recharged and native groundwater, travel times) (Clark et al., 2004, 2014; Massmann et al., 2008; Bekele et al., 2014). The flux distribution is affected by the complexity and heterogeneity of natural systems. Sedimentary deposits frequently consist of layers with varying grain size distributions that may cause the aquifer to behave locally as a multilayer system, where the actual flux distribution is not controlled as much by the hydraulic conductivity within the layers as by their continuity and
inter-connectivity, particularly in the vertical dimension (Fogg, 1986; Martin and Frind, 1998). Characterizing heterogeneity in such systems at the recharge basin scale is required for proper representation of RTDs because heterogeneity causes uncertainty (Park et al., 2006) and promotes a broad range of residence time distributions (Tompson et al., 1999). But it is hard because the head differences are small and detailed hydraulic testing difficult to perform. Even when sophisticated characterization techniques (e.g., direct push base exploration, Butler et al. (2002); Dietrich et al. (2008)) are adopted, it is cumbersome to
characterize the small scale variations of hydraulic properties.

A reasonable and easy way to address heterogeneity and RTDs is to perform tracer tests. Ironically, few tracer tests have been performed in the context of artificial recharge. Notable exceptions are the studies in Berlin, Germany (Massmann et al., 2008), which were restricted to environmental tracers due to the proximity to the water supply, and California (Clark et al., 2004; Becker et al., 2014), which used environmental and deliberate ($SF_6$) tracers. In both cases, the goal was to monitor the
recharge water plume. Both studies found a strong variation of groundwater age with depth. To the best of our knowledge, however, no test has been performed for site scale characterization. To this end, we performed a pulse injection test at the Sant Vicenç site (Barcelona, Spain) (Valhondo et al., 2014, 2015) to obtain the RTDs by monitoring breakthrough curves.

The objective of this paper is to describe the tracer test and its interpretation using both heterogeneous and homogeneous models to assess the need for model complexity, which may be required to reproduce RTDs and thus, mixing, spreading and
water quality improvement during artificial recharge.

## 2 Materials and methods

### 2.1 Site description and instrumentation

The work was performed at the recharge basin owned by the Catalan Water Agency, located at Sant Vicenç dels Horts, approximately 15 km inland from the Mediterranean shore (Fig. 1 A) along the Llobregat Lower Valley aquifer (Barcelona, Spain).

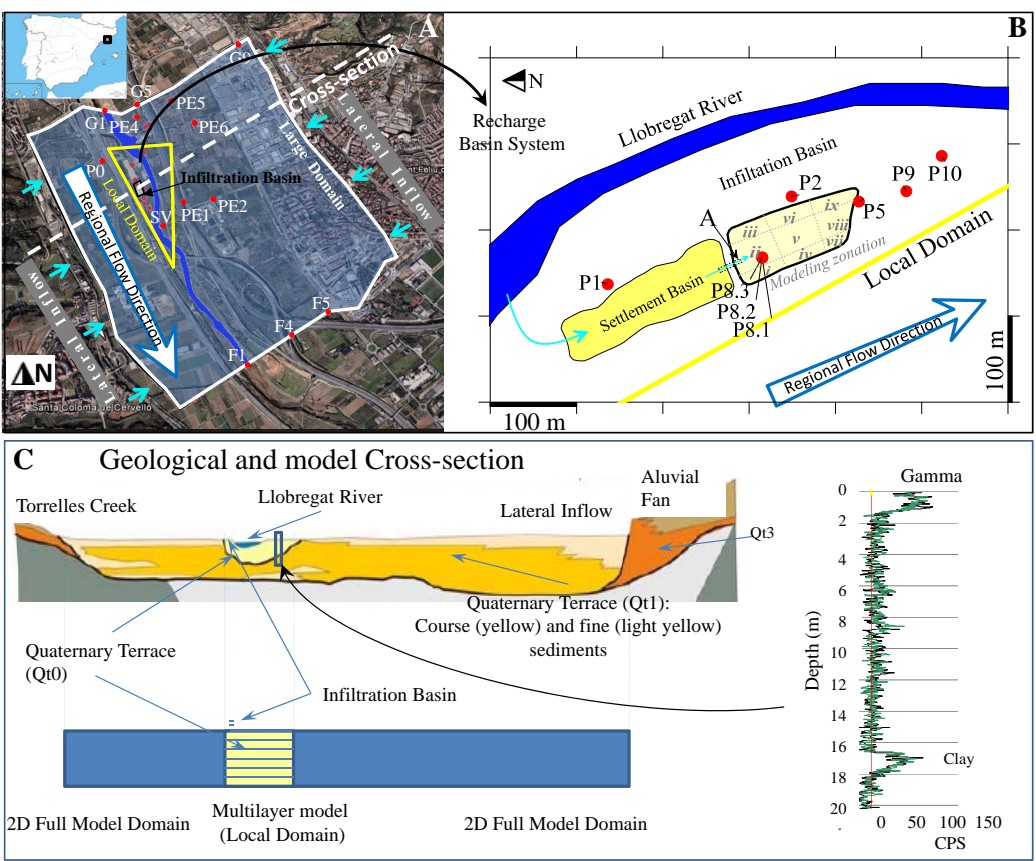

**Figure 1.** General (A) and local (B) plan views of the infiltration system, monitoring points, and model dimensions. (C) Geologic cross-section of the site and the conceptualization used in the numerical model, and natural gamma measurements from the site. The area identified with the yellow triangle in A was modeled as a multilayer aquifer (see C).

5    Recharge water is taken from the Llobregat River, which is impacted by numerous treatment plant effluents (Köck-Schulmeyer et al., 2011). River water was diverted to a settlement basin ($\approx 5000\,\mathrm{m}^2$), where it remained for 2 to 4 days. Thereafter, the water flowed to the infiltration basin ($\approx 5000\,\mathrm{m}^2$), Fig. 1 B. A flowmeter (Teledyne Isco Inc, Lincoln, Nebraska, United States) installed in the pipe connecting the two basins recorded hourly the flow rate to the infiltration basin. The infiltration rate averaged 1 m/d. The system has been operational since February 2009.

A 65 cm-thick reactive barrier was installed at the bottom of the infiltration basin in March 2011. The barrier comprises vegetable compost and aquifer sand in equal volumetric proportions and a very small fraction of clay and iron oxides. The vegetable compost aimed to release dissolved organic carbon into the infiltrating water to favor a broad range of redox conditions below the basin, thus increasing the diversity of microbial metabolic paths to enhance removal of organic contaminants (Li et al., 2013; Alidina et al., 2014; Valhondo et al., 2014, 2015). Clay and iron oxide were added to provide sorption sites for cationic and anionic organic compounds, respectively. Details on the barrier are provided by Valhondo et al. (2014, 2015).

The aquifer beneath the basin comprises Quaternary alluvial sediments with a predominant portion of gravel and sand, and a small fraction of clay (Barbieri et al., 2011). The aquifer extends to a depth of 20 to 23 m below the ground at the site and is underlain by Pliocene marls that are assumed impervious. The saturated thickness during 2010–2014 ranged from 12 to 14 m. The groundwater flows from NNW to SSE with a natural gradient of 2.3‰ (Iribar et al., 1997). Fig. 1 C shows a simplified cross-section of the site. The Llobregat River deposits are intertwined with colluvial deposits from lateral alluvial fans of local creeks, forming complex alternating layers of different compositions. The varying fractions of gravel, sand, and clay cause a significant heterogeneity in the vertical dimension (Gámez et al., 2009). This layering was represented in the numerical model as a multilayered local domain.

Eight piezometers were used for monitoring (Fig. 1). Piezometer P1, located upstream from the infiltration basin, monitored the background regional groundwater flowing under the basin. Piezometers P8.3, P8.2, and P8.1, located in the middle of the infiltration basin at depths of 7 to 9 m, 10 to 12 m and 13 to 15 m, (below the infiltration basin surface) respectively monitored the depth-related changes of the recharged water. Piezometers P2, P5, P9, and P10, fully screened and located downstream along the flow path at increasing distances from the infiltration basin, monitored the recharged water at increasing travel times. Most of the piezometers were equipped with CTD-Diver (Schlumberger water services, Delft, The Netherlands) sensors for continuous recording of electrical conductivity (EC), temperature, and water level (as pressure). The CTD-Divers were installed prior to artificial recharge. An additional CTD-Diver was placed in the middle of the infiltration basin (beside P8 nest) to measure EC, temperature, and level of the infiltration water. Samples from monitoring points and infiltration basin were collected for analysis during several campaigns (Valhondo et al., 2014, 2015).

## 2.2 Tracer test

A natural flow tracer test experiment was performed between 9 July and 14 September, 2012. Amino-G acid was selected as the tracer because it can be detected at very low concentrations, is stable in the experimental pH range (Flury and Wai, 2003), and has relatively low sorption onto organic matter and clay (Trudgill, 1987; Smart and Smith, 1976). It is, however, photo-degradable.

The recharge system was operating for three weeks before adding the tracer, while trying to maintain a 1 m column in the infiltration basin to sustain steady-state flow. On 9 July, "day 0", 8 kg of amino-G acid diluted in $0.9\,m^3$ of water from the settlement basin was poured into the entrance of the infiltration basin over approximately 15 minutes (point A in Fig. 1B) in the late afternoon to minimize photo-degradation. Breakthrough curves were measured in situ with 3 portable GGUN-FL fluorometers (Albillia Co, Neuchâtel, Switzerland) at six downstream monitoring points (P8.3, P2, P5, P8.1, P9, and P10) (Fig

1B). The fluorometers were calibrated with serial dilutions (1, 10, and 100 $\mu$g/L) prepared with the tracer and water from the settlement basin. Three fluorometers were initially installed at the monitoring points with the shortest expected travel times (P8.3, P2, and P5) and programmed to record a measurement every 5 minutes. Mean travel times were obtained from the EC recordings. Once the bulk of the breakthrough curves had been recorded at these three points, the fluorometers were moved to the monitoring points with longer travel times (P9, P10, and P8.1). Thus, one fluorometer was moved from P8.3 to P8.1 at midnight on 13 July, and the others were moved from P2 to P10 and P5 to P9 at 17:20 on 15 July. The recording interval was changed to 15 minutes on 25 July and maintained until the end of the test on 14 September.

## 2.3 Model construction

An integrated regional hydrogeologic model had been built to improve the management of water resources (Iribar et al., 1997; Abarca et al., 2006). The artificial recharge system is nested within the regional model domain. Therefore, to study the aquifer behavior in the proximity of the artificial recharge system, we created a flow and transport model based on the regional one and refined the local scale detail with information from the recharge system.

### 2.3.1 Boundary conditions and model parameterization

The model structure was defined to accommodate two requirements: (1) the need to account for layering at the local scale and (2) the need to seek appropriate boundary conditions controlling the flow field. The latter was met by choosing a (large scale) model domain bound along the flow direction by two lines of frequently measured piezometers (G1-G5-G9 and F1-F4-F5 in Fig. 1 B), $\approx$3 km away, which were used to prescribe heads at the northern and southern boundaries. The width of the model ($\approx$ 2.5 km) was established by the fluvial deposits (brown color in Fig. 1 C). Inflows from the eastern and western local creeks were prescribed using time dependent inflows of the regional model updated with recent weather data. Time dependent pumping rates were prescribed on the drinking water wells with radial galleries (PE1, PE2, PE4, PE5, and PE6 in Fig. 1 A) using data from the water utility (AGBAR). Areal recharge was prescribed by updating the regional model time functions for recharge in urban, rural, and irrigated areas. Infiltration from the river bed is small because the Llobregat River is disconnected from the aquifer most of the time (Vázquez-Suñé et al., 2006) and aquifer heads remained below the river bed elevation throughout the model period. River infiltration was taken from the regional model.

The multilayer nature of the system was modeled explicitly only in the area adjacent to the infiltration basin (local domain $\approx$0.5 x 1.5 km$^2$, yellow triangle in Fig. 1 A), where a high level of detail was needed. The rest (large domain) was modeled as two-dimensional using linear triangular elements. Therefore, the 14 m-thick aquifer was divided into seven 2 m-thick layers in the local scale domain to emulate the material differences of the alluvial deposits. Two additional 0.3 m-thick layers were implemented in the infiltration basin surface to represent retention time at the reactive barrier. The nine layers (Ly1 to Ly9 starting from the bottom) that overlapped in the local domain were linked by one-dimensional elements. The number of layers was chosen to obtain a sufficient precision in the vertical discretization while maintaining a reasonable numerical burden. Each layer was homogeneous in the horizontal direction, which is a simplification. The local and larger domains are fully coupled

and were solved together in every model run. The element size increased from 5 m at the infiltration basin to $\approx$185 m at the edges of the model.

The calibration and modeling strategy consisted of three steps. First, starting from the parameterization of the regional model by Abarca *et al.* (2006), we used updated meteorological and piezometric head data. The large scale domain was re-calibrated using the newly collected four years head data and the original transmissivity values as prior estimates. Second, we calibrate the porosity and hydraulic conductivities of the local scale domain, and the preferential flow through the reactive barrier using the piezometric heads and amino-G acid concentrations measured during the tracer test. We performed the calibration under homogeneous and heterogeneous scenarios. Dispersivity was kept constant at 50 m in the large scale domain and 5 m (longitudinal) and 1.3 m (transverse) in the local scale domain while diffusion coefficient was kept constant at $10^{-8}$ m$^2$/d with the exception of the 1D elements linking the local and the large scale domains for which the diffusion coefficient was $10^3$ m$^2$/d and the 1D elements representing wells where it was fit to $10$ m$^2$/d. Third, we validate the model by reproducing observed values of 1,1,2-trichloroethane (TCA) and EC collected under different flow conditions from those used for the calibration.

The model was built using the inversion capabilities of Transdens (Medina and Carrera, 1996, 2003; Hidalgo et al., 2004), a code that solves linear flow and transport equations for porous media, using the finite elements method in space and a weighted finite difference scheme in time, and that allows automatically estimate aquifer parameters.

### 2.3.2 Estimation of flow parameters at the large scale domain

Hydraulic parameters (hydraulic conductivity and storativity) of the local and large domain models were calibrated against the head data at the piezometers shown in Fig. 1 from January 2010 through December 2013. Several infiltration episodes took place during these four years, which were discretized into daily time steps. Prior estimates of model parameters were obtained from the regional flow model (Abarca et al., 2006) and completed with local tests (a pumping test and a convergent flow tracer test).

### 2.3.3 Tracer test and local scale parameters

The estimated flow parameters in the large scale model were used to calibrate transport parameters and recalibrate flow parameters at the local scale domain against the heads and concentrations recorded during the tracer test (from 9 July to 14 September 2012).

A 15 min wide tracer input was added to the inflow at the start of the test. The tracer was poured at the entrance of the infiltration basin (point A in Fig. 1 B) and afterwards clean water continued to flow to the basin. Thus, the tracer was not homogeneously diluted in the whole basin water volume. In fact, the maximum concentration measured at P8.3 was 2.75 times higher than the concentration assuming complete dilution of the tracer in the basin volume (1.6 mg/l). Such behavior probably reflects preferential flow through the high permeability sandy sediments below the reactive barrier as well as basin scale variability, which together with the potential photodegradation of the amino-G caused the expected distribution of the amino-G concentration to decrease from north to south in the basin. To address this issue, we divided the basin into nine zones (zones i through ix in Fig. 1 B) and estimated the concentration of the tracer at each zone as a multiple of the amino-G acid concentration

function. The effect of preferential flow, which was also apparent from the redox sensitive species (Valhondo et al., 2014, 2015), was modeled by distributing the time dependent measured inflow data between the surface of the infiltration basin in Ly9 and Ly7. Infiltration creates a downwards flux of water. Therefore, samples from the four fully penetrating piezometers should be representative of the highest permeability layer intersected by each piezometer. Since these may vary among piezometers (model layers should be understand as a conceptual abstraction), we did a preliminary screening to find which layer reproduced best observed concentrations in each piezometer.

The time discretization was quite irregular, with the shortest time steps (5 minutes) between the tracer discharge and the arrival of the breakthrough curve at P8.3 and the longest steps (3 days) at the end of the test. The standard deviation assigned to all concentration measurements at each observation point was 1% of the maximum concentration at each point, to ensure that a comparable weight was given to each point during calibration (maximum concentration varied by $\approx 2$ orders of magnitude).

The calibration yielded three models: a homogeneous one ("Hom"), where $K_x$ and $K_z$ were constant throughout the model domain, and two heterogeneous ones ("Het-1" and "Het-2") with different hydraulic conductivities for each layer. These last two models represented two different convergence points of the calibration process and both were used to highlight the non-uniqueness of the solution and to assess uncertainty.

## 2.4 Validation

The heterogeneous models have got many degrees of freedom and risk of overparameterization. This together with the inherent simplifications of the model might introduce calibration artifacts. We simulated the evolution of TCA and EC during periods of time much longer than those used for calibration to test the validity of the three models. TCA was measured sporadically in both recharge water and piezometers. We found that it was only present in native groundwater. Therefore, it could be used to test how natural groundwater recovered after infiltration periods. TCA simulations covered eight months (from April 2011 to December 2011). Initial concentration and lateral inflows of TCA concentrations were fixed as the maximum concentration observed at P1 (upstream of the infiltration basin). Artificial recharge concentration was zero. The EC simulations covered four years (2010–2013) when heads fluctuated significantly (Fig. 2). Therefore, TCA and EC test the model behavior under different flow conditions. Initial and lateral inflows of EC were fixed at 1200 $\mu$S/cm, the mean measured during the period. Recharge water EC was prescribed using a time function based on the measured EC. Both TCA and EC concentrations in the northern border of the local domain were prescribed to be equal to those measured at P1, but shifted in time for the mean travel time from the northern border to P1. Further details on the model structure and reasons behind simplifications can be found in the responses to comments by the reviewers (Clark, 2016; Walther, 2016; Anonymous, 2016; Valhondo et al., 2016a, b).

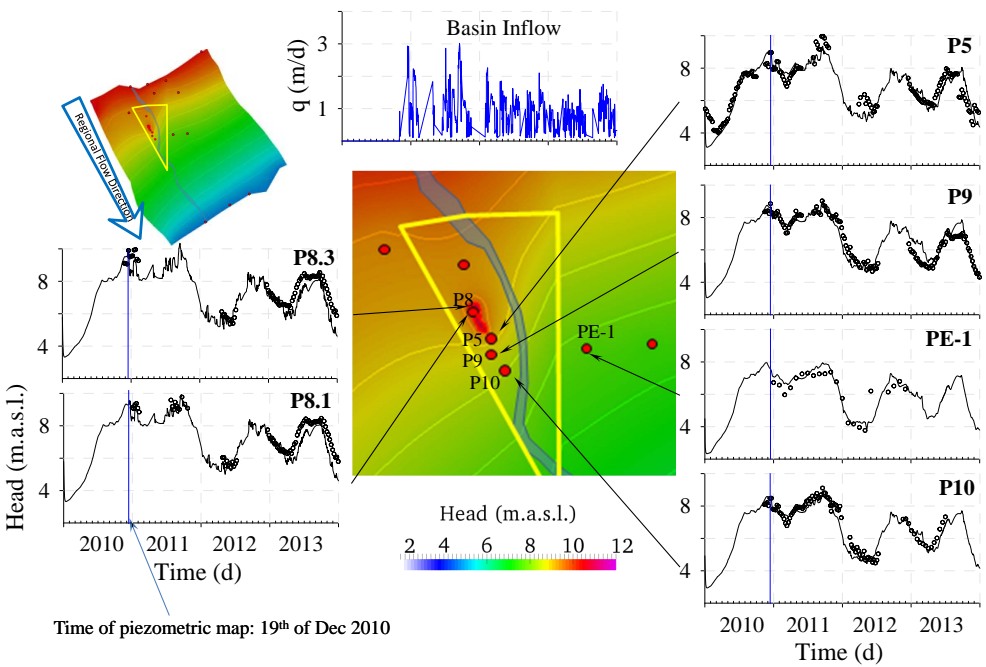

**Figure 2.** Calculated head level surface on 19 of December 2010, when the system was working, and measured (circles) and calculated (line) heads (m.a.s.l.) vs. time (m) at five monitoring piezometers (P8.3, P8.1, P5, P9, and P10) and one extraction well (PE-1).

## 3 Results and discussion

### 3.1 Flow model

Artificial recharge creates a smooth dome below the infiltration basin and modifies the piezometric surface (Fig. 2). Figure 2 displays head fits at five monitoring piezometers located within the local domain. The fit was good (mean weighted residual of

5    $-0.02 \pm 0.57$ for head observations), which suggests that the size of the multilayer local domain was sufficient to reproduce head variations at the monitoring piezometers close to the basin where the gradient is mainly vertical ($\approx 10\%$) due to the influence of the artificial recharge. The model also reproduces measurements at extraction wells outside of the multilayer local domain (e.g. PE-1 in Fig. 2), suggesting that the two-dimensional model was sufficient to create the appropriate head frame for simulating transport.

10   ### 3.2 Tracer test model

Conservative transport parameters were estimated for the nine layers of the local and basin domains using the amino-G acid tracer test data. Flow parameters of the nine layers and the one-dimensional elements linking them were also re-calibrated

because concentrations were more sensitive to vertical layering than heads, which are often only mildly sensitive to the degree of hydraulic connection (Fogg, 1986). Three sets of parameters (Het-1, Het-2, and Hom) were obtained. Estimated parameters fit, measured by the root mean square-weighted error with prior estimates (RMSWE, see Medina and Carrera (2003)), and amino-G input mass for these three outcomes are shown in Table 1.

**Table 1.** Parameters (Hydraulic conductivity, K (m/d), and porosity, $\phi$), RMSWE, and input mass in outcomes Het-1, Het-2, and Hom, after calibration.

| | | K (m/d) | | | $\phi$ | | |
|---|---|---|---|---|---|---|---|
| | | Het-1 | Het-2 | Hom | Het-1 | Het-2 | Hom |
| Basin Domain (50 x 100 m) | Ly 9 ($K_h$) | 1 | 1 | 1.4 | 0.5 | 0.5 | 0.5 |
| | Ly 9 ($K_z$) | 5.0 | 5.0 | 3.0 | 0.0001 | 0.002 | 0.0001 |
| | Ly 8 ($K_h$) | 1 | 1 | 1.4 | 0.5 | 0.5 | 0.5 |
| Local Domain (1000 x 500 m) | Ly 8 ($K_z$) | 5.0 | 5.0 | 3.0 | 0.0001 | 0.002 | 0.0001 |
| | Ly 7 ($K_h$) | 0.2 | 0.2 | 1.4 | 0.17 | 0.13 | 0.2 |
| | Ly 7 ($K_z$) | 2.4 | 3.4 | 3.0 | 0.0001 | 0.01 | 0.0001 |
| | Ly 6 ($K_h$) | 139.1 | 158.2 | 289.9 | 0.17 | 0.17 | 0.2 |
| | Ly 6 ($K_z$) | 2.4 | 3.4 | 3.0 | 0.0001 | 0.01 | 0.0001 |
| | Ly 5 ($K_h$) | 1042.1 | 1187.3 | 289.9 | 0.18 | 0.17 | 0.2 |
| | Ly 5 ($K_z$) | 93.5 | 190.5 | 3.0 | 0.0001 | 0.01 | 0.0001 |
| | Ly 4 ($K_h$) | 203.4 | 253.9 | 289.9 | 0.16 | 0.20 | 0.2 |
| | Ly 4 ($K_z$) | 42 | 210.7 | 3.0 | 0.0001 | 0.01 | 0.0001 |
| | Ly 3 ($K_h$) | 330.9 | 479.0 | 289.9 | 0.27 | 0.33 | 0.2 |
| | Ly 3 ($K_z$) | 0.2 | 0.2 | 3.0 | 0.0001 | 0.01 | 0.0001 |
| | Ly 2 ($K_h$) | 131.1 | 75.5 | 289.9 | 0.25 | 0.22 | 0.2 |
| | Ly 2 ($K_z$) | 2.1 | 29.1 | 3.0 | 0.0001 | 0.01 | 0.0001 |
| | Ly 1 ($K_h$) | 162.3 | 478.2 | 289.9 | 0.10 | 0.20 | 0.2 |
| Model Fit (RMSWE) | | 1979 | 2000 | 9358 | | | |
| Input Mass (g) | | 6683 | 6853 | 7885 | | | |

Measured and calculated breakthrough curves at monitoring piezometers are displayed in Fig. 3, which provides room for some insights. Monitoring piezometers P8.3, P2, and P5 displayed fast responses with maximum concentrations higher than those of piezometers P8.1, P9, and P10, in which dispersion and mixing generated longer tails. The monitoring point P8.3, located below the basin, was the first monitoring point reached by the tracer, and showed the highest observed maximum concentration, more than an order of magnitude higher than the next monitoring point P2. We assume that the early arrival at P8.3 occurred through preferential flow paths. The breakthrough curve at this point was very narrow, as the follow-up water without tracer reached this point equally fast. First arrival at P2 and P5 (1 day) was much faster than at P8.1 (3 days), only 6 m below the phreatic surface, despite the fact that vertical gradient ($\approx 10\%$) was much larger than the horizontal gradient (less than 4%). This observation implies that recharged water spreads laterally faster than vertically and confirms the importance of layering.

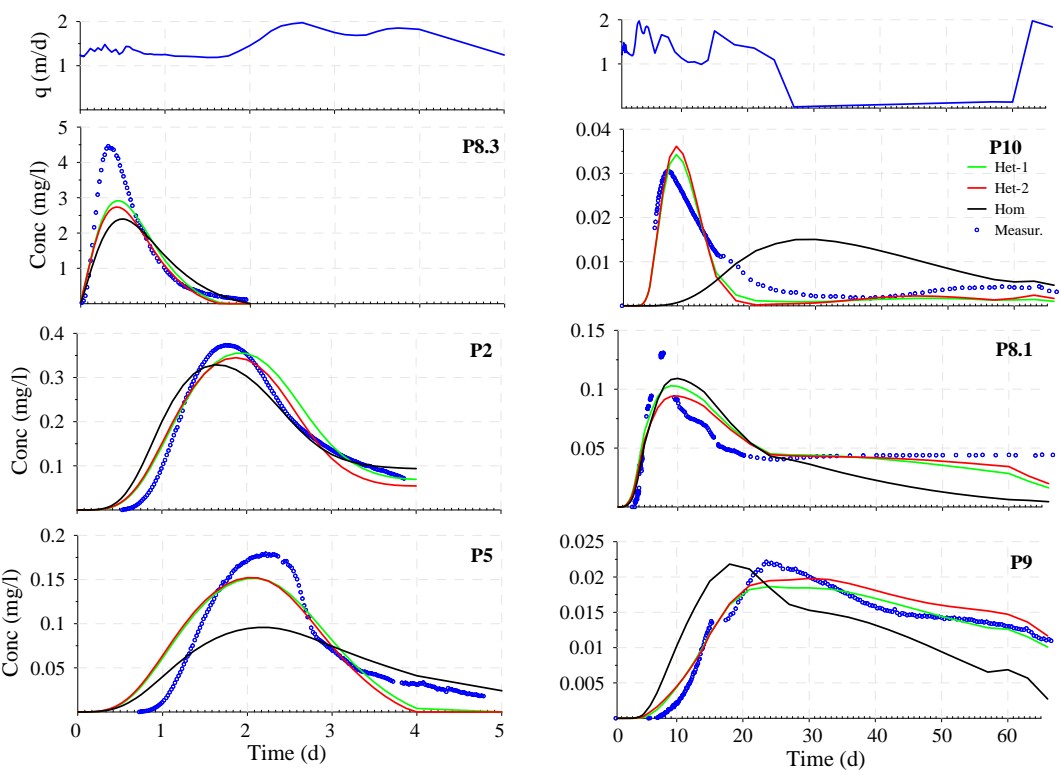

**Figure 3.** Top row, inflow rate to the infiltration basin and lower three rows measured (blue circles) and calculated breakthrough curves from the piezometers monitoring of the amino-G acid concentration (mg/l) vs. time (d) using the estimated set of parameters of Het-1 (green line), Het-2 (red line), and Hom (black line). The left row displays only the first five days of the experiment. Note that the vertical scale is different for each monitoring point.

Breakthrough curves at P10, P8.1, and P.9 exhibited longer tails than those at P8.3, P2, and P5. The short tails were consistent with the fast arrival. The long tails might suggest the impact of heterogeneity (dispersion) and mixing away from the entrance. The fact that the homogeneous model reproduced tailing quite well (at least for P8.1 and P9), however, implies that broad RTDs are caused not only by heterogeneity but also by the mean flow structure. The "shower" effect of recharge ensures that water

5 flowing initially upstream or falling on the dome top will eventually mix with recently recharged water further downstream. This effect is illustrated by the spatial distribution of the concentration shown in Fig. 4.

Several features are apparent from the spatial distribution of the concentrations. First, the distribution was balloon-like. The tracer was distributed along an outer crust that grew by filling with the tracer-less water that kept entering through the basin. Second, the portion of tracer that flowed upstream initially was eventually transported downstream through lateral

10 and downwards flow paths. This promoted shear and lateral mixing. Third, heterogeneous models provided another shear mechanism (Fig. 4 B, D and F) by the fluctuations in horizontal hydraulic conductivity ($K_x$) among the different layers. Note that the plume in layer 5 traveled much faster than in the remaining layers, to the point that it had virtually disappeared from the

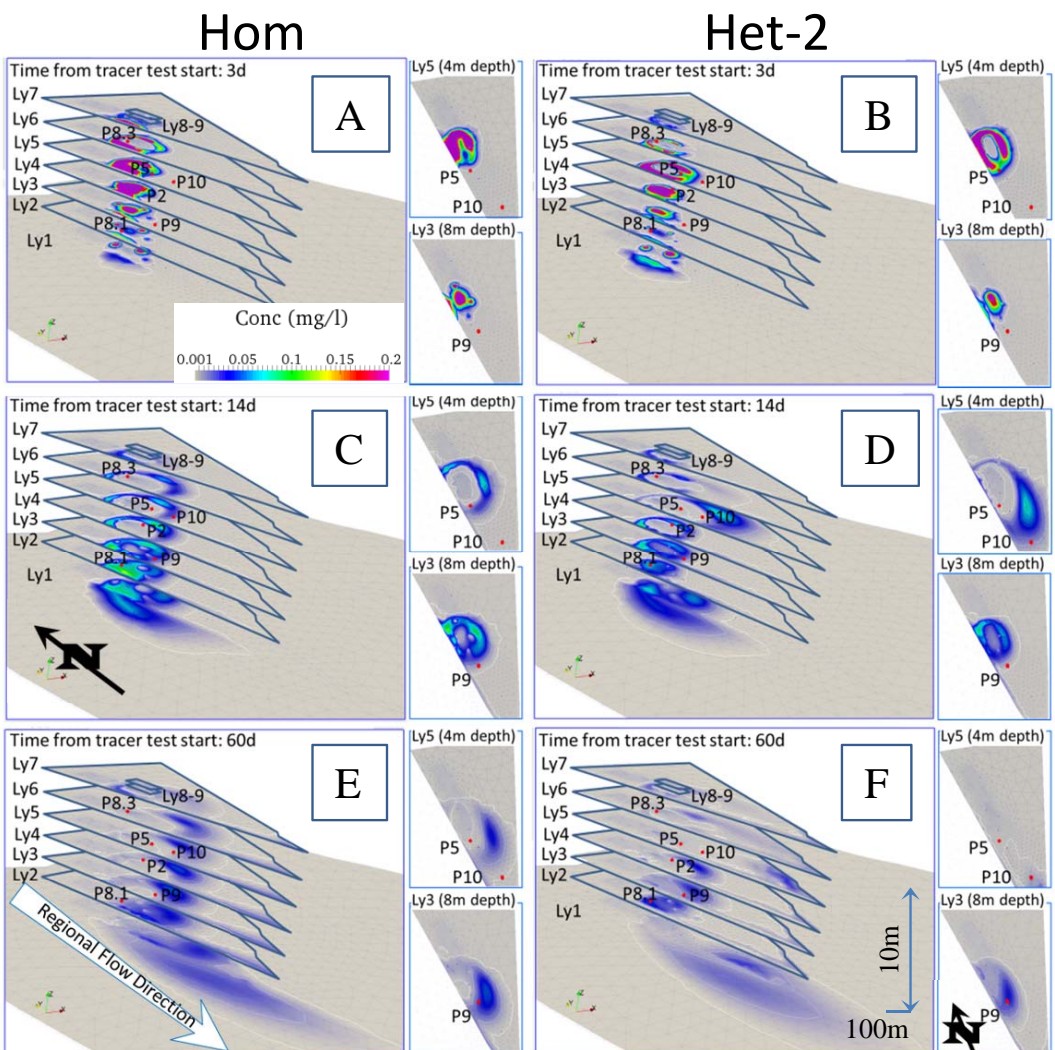

**Figure 4.** Distribution of amino-G acid concentration at three time points (3 d, 14 d, and 60 d) since the test started, calculated with the Hom (A, C, and E) and Het-2 (B, D, and F) models. The concentrations in layers 3 and 5 (Ly3 and Ly5 in the figure) are shown from below on the side of each frame. Note that the vertical scale is 100 times the horizontal scale.

local domain after 60 days. The plume also almost disappeared in Layers 6 and 7, but in this case the disappearance reflected vertical, rather than horizontal, displacement.

We contend that these shear mechanisms promoted mixing and that they were more marked in the heterogeneous models than in the homogeneous model. In fact, this observation was confirmed by the plumes shown in Fig. 4. Those of the heterogeneous models were much more diluted than those of the homogeneous model.

This kind of shear and mixing promoted broad RTDs and caused recently recharged water (possibly aerobic and loaded with dissolved organic carbon) to mix with more than 60 days old water (possibly anaerobic and depleted of dissolved organic carbon) at monitoring points P8.1, P9, and P10. Such mixing contributes to favoring the presence of a primary substrate to be metabolized by microorganisms, which increases the biotransformation of emerging contaminants, by co-metabolism. It may also explain, at least in part, the excellent performance of the system in eliminating a broad range of emerging contaminants (Valhondo et al., 2014, 2015).

Mixing at the edges of the local domain is unrealistic. As shown in Fig. 1, only the local domain was treated as a multilayer. The rest was treated as two-dimensional. This implies that, as the plume left this domain, the outflow of all the layers was mixed. This causes the plume that departs from the western edge of the local domain (plotted as Ly1 in Fig. 4) to be artificially smoothed. Whereas this mixing was an artifact of the model structure, it does not affect the computed breakthrough curves because all observation points belong to the local domain.

A final remark on the validity of the models can be drawn from the fact that monitoring point P10 was further away from the basin than monitoring point P9. Nevertheless, P10 was reached by the tracer faster than P9. Breakthrough curves of P10 and P5 were poorly reproduced under the homogeneous medium hypothesis. Both heterogeneous models, Het-1 and Het-2, reproduced the measured concentrations with better accuracy than Hom. The RMSWE values were 1979 and 2000 for Het-1 and Het-2, respectively, and 9358 for Hom. The main difference between Het-1 and Het-2 was the distribution of conductivity and porosity, because the total transmissivity of the seven layers resulted in a similar values in the three models. Model Het-2 was more consistent with the field observations regarding the materials distribution (Fig. 1 C) than model Het-1.

These observations suggest that the Het models were better than the Hom model. But they may be overparameterized (Poeter and Hill, 1997; Carrera et al., 2005). It is clear that the heterogeneity assumption is required to reproduce geologic observations, which is valuable information in itself (D'Agnese et al., 1999), and to model mixing (Le Borgne et al., 2010). It is also clear, however, that parameterizing heterogeneity causes non-uniqueness. In fact, the fast arrival at P10, which we reproduced by the high hydraulic conductivity in layer 5, might reflect other causes (e.g., a high-permeability paleochannel within layer 5). Therefore, it would be fair to question the validity of explicitly modeling heterogeneity. We address this question below.

## 3.3 TCA and EC validation

The validity of the models calibrated with the tracer test, as discussed in section 3.2 above, was tested against measurements of TCA and EC. Modeling these simply required changing initial and boundary concentrations (see section 2.4).

Figure 5 displays the changes in the measured and calculated concentrations of TCA ($\mu$g/L) at the monitoring piezometers. Measured TCA concentrations approached the background concentration of the aquifer (some hundreds of $\mu$g/L but varying) when the artificial recharge system was not operating. Concentrations of TCA decreased and fell below the detection limits at most monitoring points when the recharge with TCA-free water was activated. These trends were generally reproduced by the three models and confirmed that the observation points sampled recharge water. The models were far slower in reacting to changes in recharge rate, however, than the actual observations. In particular, they were too slow to reproduce the TCA

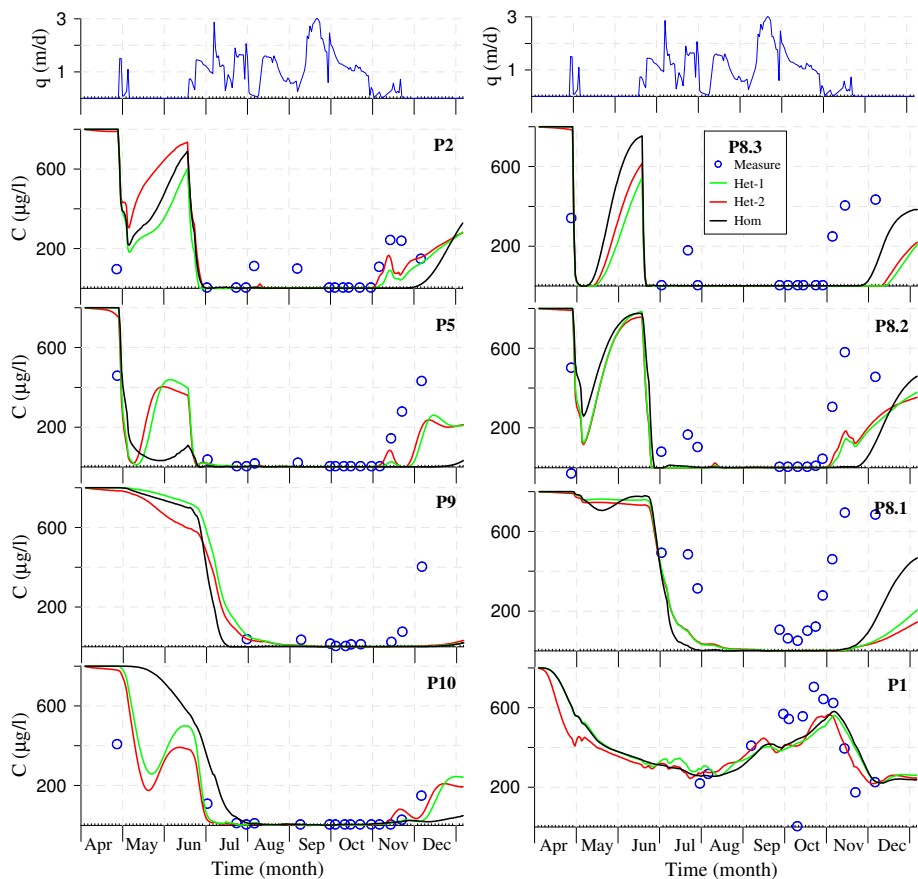

**Figure 5.** Measured (dark blue circles) and calculated TCA concentration ($\mu$g/l) changes over time at monitoring piezometers for the Het-1 (green line), Het-2 (red line) and Hom (black line) models. The infiltration rate is also shown (top).

concentration rebound after the recharge stopped in Nov-Dec. Still, with the exception of P8.3 and P8.1, heterogeneous models did a better job than the homogeneous one, and Het-2 performed slightly better than Het-1.

Figure 6 displays the changes in measured and calculated EC ($\mu$S/cm) during two years. The three models reproduced the observations quite accurately, except during the low recharge period at the end of 2012 and the beginning of 2013. Measured
5    EC at P2, P5, and P10 during this period fell below both the recharge water EC (similar to that of P8.3) and aquifer water EC (similar to P1). Therefore, the error must be attributed to some unaccounted inflow of low EC water rather than to poor model structure.

In summary, the three models reproduced quite well the change in TCA, which was present in the aquifer but not in the recharge water, and EC, which fluctuates in both. On the one hand, this implies that the velocity field, imposed by recharge
10    and natural aquifer flow, was not overly sensitive to local hydraulic conductivities. On the other hand, it implies that it would be difficult to accurately estimate the layering structure solely based on the concentration breakthrough curves. In fact, the

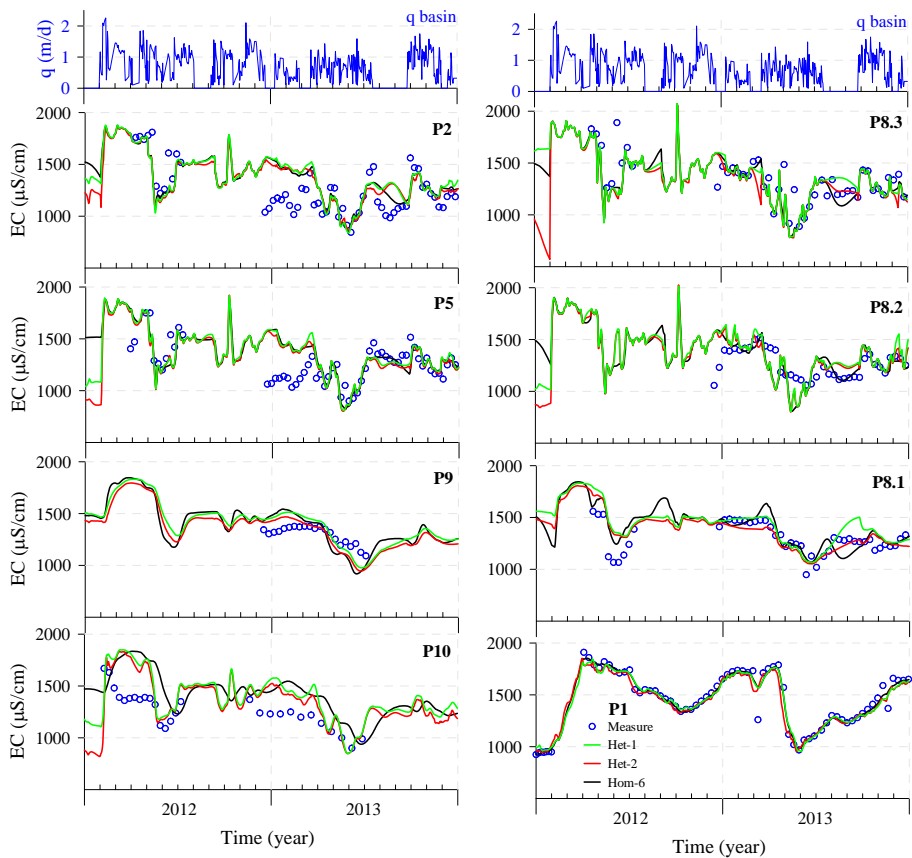

**Figure 6.** Changes in measured (dark blue circles) and calculated EC ($\mu$S/cm) at monitoring piezometers for the Het-1 (green line), Het-2 (red line) and Hom (black line) models. The infiltration rate is also shown (top).

RMSWE for EC with the Hom model (4629) was slightly smaller than that for the Het-1 and Het-2 models (4682 and 4689, respectively), which suggests that the Hom model, having less uncertain parameters, was more robust than the Het models, at least for EC.

One might be tempted to use multicontinua models (Haggerty and Gorelick, 1995), which reproduce the effect of hetero-
5   geneity (Silva et al., 2009; Dentz et al., 2011). In fact, such models would probably capture the fast rebound of the TCA concentration when the recharge stopped, as observed in Fig. 5. In view of the calibration non-uniqueness of these models, however, would probably worsen identifiability.

## 4   Conclusions

This work provides useful insight on both tracer testing for characterization of artificial recharge and on transport modeling.

The tracer test was successful in identifying RTDs at a number of piezometers. These distributions were quite narrow at points immediately adjacent to the basin (P8.3, P2, P5) but were very broad (more than 60 days) at points slightly further away (P8.1, P9, P10). Broad RTDs imply significant mixing of recently recharged water with water recharged some time before. Such mixing, together with the conditions imposed by the reactive layer, promote diverse metabolic paths and helps to explain the effective removal of a wide range of emerging contaminants at this site (Valhondo et al., 2014, 2015).

The model suggests that the broad RTDs are the result of both the flow structure, which is complex, and heterogeneity. Recharged water flows initially upstream and then laterally around and below recently recharged water. This complexity stretches flow tubes and favors mixing (Dentz et al., 2011). This effect is enhanced by temporal fluctuations in recharge and was observed in both the homogeneous and heterogeneous models. Further shear, stretching, and mixing was caused by the variability in the hydraulic conductivity among layers.

Regarding transport modeling, it is clear that the collected breakthrough curves were not sufficient to identify the hydraulic conductivities of the modeled layers or, even less, more complex heterogeneous structures. In fact the simple homogeneous model, which did not perform as well as the two heterogeneous models during the tracer test calibration, yielded similar (if not better) blind predictions of EC under varying flow conditions, and only slightly poorer for TCA.

The introduction of heterogeneity is justified not by the quantitative head or concentration data, but by geologic understanding. Ultimately, the actual RTDs that are required for proper interpretation of pollutant removal, were well reproduced by the heterogeneous models. Therefore, these models should be used for interpreting and predicting the fate of recharge water. Yet, given the importance of artificial recharge and that its clean-up potential can be enhanced by time fluctuations of recharge rate (de Dreuzy et al., 2012), much can be gained by the detailed characterization of recharge sites. To this end, tracer tests are useful, but insufficient. They must be complemented with cross-hole inter-layer testing and other techniques (e.g., geophysics, direct-push tests).

*Author contributions.* All authors have contributed to the work. Jesús Carrera, Carlos Ayora, and Cristina Valhondo designed the tracer test experiment and the three of them together with Katrien De Pourcq and Alba Grau-Martínez carried it out. Jesús Carrera and Juan J. Hidalgo developed the model code, and advised Cristina Valhondo, Lurdes Martínez-Landa, and Isabel Tubau performing simulations and calibrating flow and transport parameters. Cristina Valhondo and Lurdes Matínez-Landa prepared the manuscript counseled by Jesús Carrera.

*Acknowledgements.* Most of the work was supported by the ENSAT project funded by the LIFE program of the European Commission (ENV/1225 E/117). Final portions were performed with funding from EU project ACWAPUR (PCIN-2015-245) and the European Research Council through the project MHetScale (FP7-IDEAS-ERC-617511). The authors would like to acknowledge the Geo-sciences department of the University of Rennes for supplying the fluorometers, AGBAR and Catalan Water Agency for collecting and sharing the hydro-geologic data, and the Hydrogeology Group (UPC-CSIC) for it implication in the tracer test performance. The authors thank Dr. Jordan Clark, Prof. Marc Walther and one anonymous reviewer for their constructive comments which helped us to improve the manuscript.

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
