# Peer review of "Tracer test modeling for characterizing heterogeneity and local scale residence time distribution in an artificial recharge site"

_Hydrology and Earth System Sciences, 2016_

## Referee Comment (RC1) · J. Clark (Referee) · 14 Jun 2016

Attached is my Discussion of the Valhondo et al. paper. I also found a few typos in the paper that you may want to fix:

P. 2, line 23: Becker et al. (2015) completed their work at the San Gabriel Spreading Grounds Test Basin in Los Angeles County, CA not in Orange County, CA as stated in the text. The paper was published in 2014 not 2015.

P. 7, line 1 (below figure): Typo. "Figure2" should be "Figure 2"

P. 11, line 7: Typo. "iwas" should be "was"

P. 15, line 16: Typo. "10B" should be "10B" (10 needs to be superscripted).

P. 15, line 17: Typo. "2015" should be "2014".

Please also note the supplement to this comment:
http://www.hydrol-earth-syst-sci-discuss.net/hess-2016-197/hess-2016-197-RC1-supplement.pdf

———————————————————

[Figure]

**Supplement:**

Discussion of "Tracer test modeling for local scale residence time distribution characterization in an artificial recharge site" by Valhondo et al.
(doi:10.5194/hess-2016-197)

Jordan F. Clark
jfclark@geol.ucsb.edu
Dept. of Earth Science, University of California, Santa Barbara, CA 93106 USA

**Commentary**

Valhondo et al. (2016) is an important paper that exams near-field flow under managed aquifer recharge (MAR) spreading ponds. It combines both geochemical tracer techniques and numerical modeling of flow and transport. The combination of these separate approaches reveals the complexity of flow beneath spreading ponds. I suspect that this is largely due to the local hydrogeology. Spreading ponds are often located in alluvial settings such as the one examined near Barcelona, Spain by the authors.

A unique contribution of this paper is the careful consideration given to the vertical structure of the aquifer below the spreading pond. The authors created a finer scale grid with nine layers that was imbedded into a regional numerical model of flow by Abarca et al. (2006). Three different fine scale models were tested: "Hom"—containing homogeneous $K_h$ and $K_z$; "Het-1" —containing different $K_h$ for each of the nine layers while maintaining the same $K_z$ for each layer; and "Het-2"—containing different $K_h$ and $K_z$ for each of the nine layers (see Table 1 of the paper). Field measurements of head and geochemistry were used to validate the fine grid model. I particularly appreciated the use of TCA (1,1,2-trichloroethane) as way to quantify the advection of regional groundwater into the study area.

The most significant contribution of Valhondo et al. (2016) is their characterization of preferential flow in the heterogeneous aquifer found in their study area. Unfortunately, they did not discuss the work of Thompson et al. (1999) who produced one of the original tracer data numerical flow models to interpret the complexity of flow and transport near MAR sites. They used a different approach but reached a similar conclusion. As mentioned above, MAR sites are more often than not located above heterogeneous aquifers, so the authors' findings should be applicable to other settings. As Fox et al. (2007) demonstrated many years ago at the 6[th] International Symposium on Managed Aquifer Recharge (ISMAR6), the placement of monitoring wells for management purposes must account for preferential flow. Without using complex numeral model such as those employed by Thompson et al. (1999), Fox et al. (2007) and Valhondo et al. (2016) or detailed deliberate (added) tracer experiments (e.g., Clark et al., 2014; Becker et al. 2014), it is hard to demonstrate the residence time distribution and hydraulic connection between the recharge area and monitoring well. Therefore documentation of water quality changes is uncertain and must be recognized.

References

Abarca, E., Vázquez-Suñé, E., Carrera, J. Capino, B., Gámez, D. and Batle, F: Optimal design of measures to crrect seawater intrusion, Water Resources Research, n/a-n/a, doi:10.1029/2005WR004524, w09415, 2006

Becker, T. E., Clark, J. F. and Johnson J. A.: [10]B-enriched boric acid, bromide, and heat as tracers of recycled groundwater flow near managed aquifer recharge operation. *J. Hydrol. Eng., ASCE,* doi:10.1061/(ASCE)HE.1943-5584, 2014.

Clark, J. F., Hudson, G. B., Davisson, M. L.; Woodside, G. and Herndon, R.: Geochemical imaging of flow near an artificial recharge facility, Orange County, CA. Ground Water, 42, 167-174, 2004.

Fox. P., Park, H., and Cha, D-H: Uncertainty analysis of mound monitoring for recharge water from surface spreading basins, in: Management of Aquifer Recharge for Sustainability, ISMAR6 Proceedings, edited by Fax, P., Acacia Publishing, Phoenix, AR, USA, 423-432, 2007.

Thomson, A. F. B., Carle, S. F., Rosenberg, N. D. and Maxwell, R. M.: Analysis of groundwater migration from artificial recharge in a large urban aquifer: A simulation perspective. Water Resources Research, 35, 2981-2998, 1999.

Valhondo, C, L. Martinez-Landa, J. Carrera, J. J. Hidalgo, I Tubau, K. De Pourcq, A. Grau-Martinez, C. Ayora (2016) Tracer test modeling for local scale residence time distribution characterization in an artificial recharge site. *Hydrol. Earth System Sic. Discuss.,* doi:10.5194/hess-2016-197.

---

## Referee Comment (RC2) · M. Walther (Referee) · 29 Jun 2016

Comments on "Tracer test modeling for local scale residence time distribution characterization in an artificial recharge site" submitted to HESS by Valhondo et al.

The manuscript features investigations on a tracer test in an artificial recharge site by utilizing a simulation approach. Different implementations of the study area are realized with homogeneous and heterogeneous hydrogeological setups. The motivation is to understand the relevance to implement different (heterogeneous) layers in order to represent correct flow and mixing behaviour of juvenile and upstream recharge waters to evaluate the performance of AR for contaminant removal. A key factor for this is the residence time distribution which was measured in a field campaign by breakthrough curves of a tracer test. The submitted manuscript is structured in a logical way, giving a comprehensive introduction and motivation, before presenting materials and methods, showing and discussing results, and finally, drawing some short conclusions. The work is of high quality, written in a clear and understandable way, while some figures and tables support the text, and relevant references are cited appropriately. The authors manage to resemble the measurements with the model software remarkably well, albeit the relatively complex study area. By comparing results from homogeneous and heterogeneous setups, conclusions clearly show deficits of a homogeneous setup. Yet, still some questions remain open for further investigation (e.g. uniqueness of two calibrated heterogeneous setups). Already with these two aspects, I think that the manuscript is of high relevance for current research. Nevertheless, the manuscript also can be improved, especially in the following major points: a) The description and motivation of the used "tracers" (amino-G, TCE, EC) should be given in a clearer way for the reader to understand which is used for what purpose. This may be done by giving a short overview in the beginning of the methods section. b) The information given on the modeling tool are too scarce. A very short description on the type of the tool, and its features should help to understand the decision to use this tool. Furthermore, the description on the modeling strategy, and the implementation of the model (model setup) could need some more structuring (clear description of all boundary conditions, section 2.3, and full list of calibrated parameters). Also, information on the calibration strategy should be provided. c) Finally, I would like to encourage the authors to state a more profound argumentation why they set up the heterogenous models in the way they did. For example, why were 9 layers chosen and not 5, 15, or 40? I think that this can aid to support their conclusions, ie. to highlight relevance of a layered model structure in the study area.

With these general comments, I enclosed a list of more specific ones. If the authors decide to reply to these comments, I would be available for a second revision, if the authors think that my comments gave a valuable input.

[Figure]

I think, that this manuscript is already of high quality and tackles an important topic. I, therefore, suggest to consider the submission for publication after revision.

Abstract: Page 2, Line 8: You say that you used the numerical model to "...extend characterization to other flow conditions...", but I could not find any scenario simulations that are different from the observation times in the manuscript. Could you please rephrase this?

Page 3, Line 24: You write "Characterizing heterogeneity in such systems at the recharge basin scale is required for proper representation of RTDs. But it is hard because the head differences are small and detailed hydraulic testing difficult to perform." - While the mentioned issues are problematic indeed (ie. difficulties with small head differences, and to get detailed information of hydraulic properties), I would also add, that measuring hydraulic properties and the distribution of the heterogeneity (micro and makro scale) is unfeasible, as you would never be able to cover the whole basin in the full resolution.

P4, L7: "The objective of this paper is to describe the tracer test and its interpretation using both heterogeneous and homogeneous models to assess the need for model complexity." - The objective of paper you describe here is somewhat different from the one you give in the title: here, it sounds as if you wanted to evaluate the necessity of complexity in a numerical model, while the title leads to the intention that you want to model RTD in an AR site. Can you please synchronize these locations (ie. add the missing part in the text or rephrase the title)?

Figure 1: * (C) - Can you please add the location of the cross-section in (A) and/or (B)? - I do not understand what the blue arrows indicate - can you please explain these? - What are "thick" and "fine" sediments? - Can you please add a legend to the statigraphic log (or add the description of the layers right next to it)?

P6, L2: You write that "The Llobregat River was disconnected from the aquifer...", but the arrow in fig 1B shows a groundwater flow direction parallel to the river; can you

please explain how you came to the conclusion of a disconnected river?

P6, L3: A question out of curiosity on the operation of the AR: what is the reason to divert the river water firstly to a "settlement basin" - is this due to high turbidity of the river water? Can you please provide an estimate of the volume of the basins (you only give an area here)?

P6, L5: How was the flow rate measured?

P7, L6: By writing "depths of 7 m" etc, you mean "meter below ground surface" or "meter below groundwater level"?

P7, L7: Can you please tell in which layers or depth the piezometers P2, 5, 9, and 10 are filtered?

P7, L12: Can you please add the location of the addition CTD-diver to the map (fig 1)?

I like the detailed description of the measuring campaign; this provides a solid information for the interpretation of the results!

Seciont 2.3.1 needs to be clearer. In the first paragraph, it is not obvious, which combination of boundary conditions you used to acquire information for the local domain BCs, and which type of BCs you implemented. Eg: P8, L23: You describe that you added inflow from "eastern and western local creeks". - I cannot find them in figure 1; can you please consider adding them in figure 1 for a better overview? - I assume, you used a Neumann-type boundary condition for implementing the creeks (as you write "inflow"). What type of BC did you use in the regional model? Did you also include the abstraction of the drinking water wells in the regional model to get the values for the creeks' inflow? - Why did you model the eastern part from the Llobregat river? Is this part of the domain relevant for the western model domain?

Why did you choose a triangular local model area? Wouldn't an area that is limited on the lateral sides by flow paths that provide a no-flow boundary be a better solution?

P9, L4: Add comma in "(local domain ≈0.5 x 1.5 km2, yellow triangle in Figure 1 A),"

Section 2.3.2 is a little confusing: I expected that you talk about the model setup of the local scale model (yellow triangle in fig 1), as you motivated the setup of this smaller model from an existing regional scale model in 2.3. Section 2.3.1 then gives BCs etc for the small scale model, as 2.3.3 give more details on it. In between, section 2.3.2 feels a little lost. Maybe, the information you provide there (hydraulic parameters, head data), could be given before 2.3.1? Besides this, the heading of 2.3.2 does only suspect info on the large scale model, while there is actually also some info on the local domain.

P9, L8: You say that there is a strong local heterogeneity (layering due to fluvial deposition). Can you please motivate your decision to use exactly 9 layers? How many hydrogeological layers can you find in the domain? Do these 9 model layers represent major (important) hydrogeological layers at the site? What would happen, if you used 100 layers (ie. using more than one grid layer per hydrogeol. layer)? Can these 9 layers represent vertical distribution of a tracer within one hydrogeol. layer?

P9, L8: You write, that the nine layers "overlapped in the local domain". Do you mean vertically overlapping of the elements (ie. you get prism elements) or elements overlapping in the way of "being at the same location"?

P9, L9: You state, that you "linked" the layers "by one-dimensional elements". - What is the motivation to do so? Are the layers you introduced in the 3d part of the model domain disconnected initially? Do these linking elements exist for the communication of the layers? - Did you link at nodes or elements? - How many 1d elements did you use to link two layer's nodes/elements (only one line)?

P9, L10: You describe, that you introduced two top layers (8 & 9) for the reactive barrier. Assuming, that the preferential flow direction is vertical, and considering that you use linear elements for vertical exchange, why is it necessary to use a second layer for this feature - wouldn't one be enough? Besides that, can you say, that this top reactive barrier layer is having homogeneous properties in horizontal extent? Could it be clogged

preferably at the inflow zone due to stronger biological activity and availability of nutrients? (Later, you also distribute the tracer along the infiltration bed, based on a similar assumption.) What would be the influence of such reactive barrier heterogeneity on the distribution of the infiltrating water/tracer?

P9, L15: I do not understand, why you used the TCA for validation - I thought that you wanted to use the amino-G tracer for this? How did you probe for and measure TCA? Some TCA information is partly given in 2.4, but at this section it is not clear to me, why TCA is used. Maybe, you can write a short overview-like, introductory information on the intended usage of TCA and amino-G in 2.3 (or even 2.2)?

P9, L17: What is "Transdens"? A modeling tool? Can you please give a little more information on this?

P10, L9: "...whole basin water volume, as a consequence the maximum..." - I suggest to divide this sentence (and add a comma): "...whole basin water volume. As a consequence, the maximum..."

P10, L6 - L15: The reason to divide the basin into nine zones was hard to understand in the first place, as you state some results where the reader expects the model setup (which it indeed still is). Yet, there are some questions: What is the "amino-G acid concentration function"? Is this the concentration you measured at P8.3 (probably not)? Or is it defined through the input of amino-G at point A? How did you measure this function? Also, it is unclear to me, how you defined the different zones i-ix and on what basis you set values for the distribution parameters within the zones? Can you plot or state the distribution parameters in a table?

P10, L16: I don't think it is important to state that "time discretization was quite irregular". It is common to have a variable time stepping in a numerical model. The relevant questions for time step sizes would be, how you defined the different time step sizes (usually based on the error of an iteration, but here probably defined through the measurement time steps?), and whether large time steps still yield valid Courant numbers

(if applicable for your model).

P10, L21: Ly = Layer? Why didn't you distribute the inflow also on Layer 8? Did you distribute the inflow equally or weighted?

P10, L22: "The standard deviation assigned to all concentration measurements at each observation point was 1% of the maximum concentration at each point, which was to ensure that a comparable weight was given to each point during calibration (maximum concentration varied 25 by $\approx$2 orders of magnitude)" - I think this is an unfortunate decision. With this assumption, if an observation point has a low concentration, it must show a much more accurate measurement than a high concentration point. Thus, the points would be biased and low concentration points would be forced to perform much better. Wouldn't it have been better to use the standard deviations from the calibration/regression line of the fluoresence meters?

P11, L1: "...two different convergence points of the calibration process..." - Where there more than those two convergence points? - You say, that you defined two heterogeneous property distributions from the calibration. Could it also be, that the conceptual model has a fault, if the calibration does not produce a unique result? Did you consider other setups?

P11, 2.4 Validation: - At this point of the manuscript, I do not understand, how you use TCA and EC data for validation. Do you try to calibrate porosity, dispersivity, retention or decay coefficients? Why do you need two parameters for this? Maybe, you can give a short, comprised overview of what tracers you use for what purpose in the introduction? - Because of the first point here, it is also hard to understand the usage of the boundary conditions: + Why do you set TCA to the maximum value but EC to the mean? + Why do you use measured EC time series as the recharge BC? + What are the "lateral inflows"? Can you show that in fig 1? + Did you simulate decay of the TCA? + I am sorry, but I do not understand this sentence: "TCA and EC concentrations in the northern border of the local domain were prescribed based on the TCA and EC

measured at P1 and adjusted for the travel time from the northern border to P1."

Figure 2: - Time unit of head series is probably not days (looks like months with years as axis).

P13, L2: "...surface (Fig. 2). Figure2 displays..." - Please add a space in Figure2. - I think, you should also decide on using either "Fig." or "Figure".

P13, L2: "...piezometers located around local domain." - I think, it was better to write something like "...piezometers located around the local domain.", or even "...piezometers located within the local domain."

P13, L3: "The fit was good,..." - Can you please give a quantitative measure (regression coefficient, RMSE, Nash-Sutcliff or similar)?

P13, L4: "The fit was good, which suggests that the size of the multilayer local domain was sufficient to reproduce head variations at the monitoring piezometers close to the basin where the gradient is mainly vertical ($\approx$ 10%)." - Before, you said, that you produced three model variations; which one is used here to show fig 2? I would also argue, that if you have more than one unique solution (if the amount of measured data is enough), a good fit does not necessarily mean a sufficient model setup. - A gradient of 10% is a large value (10 cm head difference in 1 m, understandably for the AR), but not "mainly vertical". I suggest to rephrase this sentence to highlight the influence of the AR on the head distribution. - After reading explanations on fig 3, I undertand, that you are referring to the vertical gradient itself. Please clarify this.

P14, L2: "Conservative transport parameters were estimated for the nine layers of the local and basin domains using the tracer test data." - I would say, that this sentence should be given somewhere in 2.2 or 2.3, as it is more related to methodology than results. - Which "tracer test data" do you mean - TCA, EC or amino-G? - You never give a full list of transport parameters, please add this.

Table 1: - What is the thickness of the layers? Can you add this to the table? Then, you

could also easily compare and evaluate the results to the hydrogeological and gamma logs you give before. - Layer 5 shows a hydraulic conductivity of ∼1000 m/d, which is ∼1e-2 m/s, a relatively high value - can you justify this (and the other values) from the hydrogeology? I cannot evaluate it, as you do not give exact information on the sediments (apart from being fluviatile). - Values for the reactive barrier seem relatively high, while I would expect them to be lower due to the finer material, clogging etc. Can you please explain this? - Also, in the reactive barrier layers, vertical conductivity is higher than horizontal - why? - Why are vertical porosity values so extremely low? Can you give an explanation within a physical meaning for such small values? - Do I understand it correctly, Layers 9 and 8 are only present at the infiltration basin? If yes, what happens to the vertical elements of Layer 7 where there is no layer above? - What is the meaning of the RMSWE values for hydraulic conductivity? I know, how to get RMS(W)E for hydraulic head or concentrations (or whatever primary variable you calculate) but not for a model parameter, if you do not have otherwise measured reference values. Can you please explain this, together with how you weighted the values? - Where is the "input mass in outcomes Het-1, Het-2, and Hom, after calibration"?

Figure 3: - Why does the tracer arrive earlier at P10 than P9, although the first lies behind the latter? I would expect the opposite. (You mention that in P19, L26ff, but do not explain the reason or state a possible explanation.) - It is hard to compare the breakthrough curves, as I could not find information on the filtered area of P2-P10. Please add this information. - Can you explain the unusual tailing of the measurements in P5? - Did you consider effects from double porosity to explain the long tailing of P8.1 and P9?

Figure 4: - The grey areas, should most likely show concentrations of ∼0 mg/l, but the legend does not show a grey value. I guess that this is due to lighting settings in Paraview (looks as if you were using this software to produce the figures). Can you please check this?

Figure 5: - From which year is the data you show here? - The legend says "Homo", in

the text you defined "Hom" to be the homogeneous case.

P20, L4: "...because the aquifer transmissivity in the local domain was ultimately the same." - I do not understand this; I thought, that your model had the same setup (i.e. layers have same size); now, if you use different hydr. conductivities, you should end up with different transmissivities. Can you please explain this (I am probably misunderstanding this)?

Questions on the model: - Do you define layer thickness through transmissivity of the layers? - What happens, when a solute in a top layer reaches the boundary of the local domain? Is it moving thÅȚough the outermost vertical elements to the lower layer? Does this vertical travel increase spreading of the plume? - Can you please provide a full list of parameters (eg dispersivity, diffusion coefficients...)?

Generell questions: - What is the distance to the groundwater table? I am asking to understand whether it would be important to include the passage through the unsaturated zone. - What was your calibration strategy? Did you conduct a manual or automatic calibration? Which parameters did you use for calibration (TCE, EC, amino-G, heads) and what was the target function (RMSE)? This is important to know in order to evaluate, for example, whether there could actually be more than two fitting calibrations (currently het-1 & het-2), and whether the current conceptual setup may be the reason for non-uniqueness. - The heterogeneous setup clearly shows a better performance (at least for the amino-G tracer). To what extent, do you think, should this heterogeneity be reproduced in the model? Would you think that 3 or 300 layers showed similar results? How could one determine the necessary degree of heterogeneity (or the "sensitivity" of heterogeneity)?
* * *

---

## Author Comment (AC2) · 7 Jul 2016

Reply to comments by Prof. Marc Walther

1. Introduction

We thank Prof. Marc Walther (and will also do it in the acknowledgement section of the revised version of the paper) both for his kind assessment of our work and for the time he has devoted to improving the paper, as can be derived from the length of his comments. In the spirit of HESS discussions, we discuss below the issues that are potentially controversial or that require further explanations. Editorial corrections will be included in the revised manuscript, when a full response to all reviewer comments

will be produced but are not addressed here. Overall, the reviewer raises a number of issues that are critical for a good model, but are hard to explain in detail in the limited length of a paper. Since HESSD remains open, we will use this response to clarify many of those issues, which are relevant but will not be included in the revised version of the paper for lack of room, as a sort of "supporting information". This implies that, if Prof. Marc Walther does not mind, we will cite his comments (Walther, 2016) and our responses in the final manuscript. We also have taken the freedom to alter the order of the comments to facilitate responding in a coherent way. Comments are structured around three "major points: a) The description and motivation of the used "tracers" (amino-G, TCE, EC) ... b) The information given on the modeling tool ... the modeling strategy and ... information on the calibration strategy should be provided. c) Finally, I would like to encourage the authors to state a more profound argumentation why they set up the heterogenous models in the way they did. For example, why were 9 layers chosen and not 5, 15, or 40?". We structure this response around these three points, starting from the second one.

2. Modeling tool and strategy. Calibration strategy.

The modeling tool used for calibration is TRANSDENS (Hidalgo et al.,2004 ), which is a development over TRANSIN (Medina and Carrera, 1996; Medina and Carrera, 2003). Both codes have been developed by the Hydrogeology Group (UPC-CSIC). Both use the Finite Element Method to solve the equations governing coupled flow and transport through porous media. A strict Galerkin method is used for transport, which places strong constraints on the adopted dispersivity, but which was not an issue in this case because of the strong heterogeneity. The singularity of these codes lies on their versatility to accommodate geology, zonation, time dependence of aquifer parameters and model inputs, and especially inverse modeling. That is, they allow automatic calibration of aquifer parameters (transmissivity, storativity, recharge, boundary heads and flows, leakage, dispersivity, molecular diffusion, porosity, retardation, linear decay, boundary concentrations) using the methods described by Medina and Carrera (1996) and Medina and Carrera (2003). A short explanation will be included in the revised document in the section 2.3.1 "Boundary conditions and model parameterization". The model structure was defined to accommodate two requirements: (1) the need for detailed description of layers at the (local) transport scale and (2) the need to seek appropriate boundary conditions controlling the flow field. Overall, our large scale model is a portion the regional model of the Llobregat delta main aquifer (Abarca et al., 2006). The regional model boundaries were the natural edges of the Llobregat delta main aquifer. Our model, which is limited to a small portion of the Llobregat River alluvial aquifer, increases the detail at the local scale (around the infiltration system). Therefore, we distinguish two domains:

(1) A single layer large scale domain in the model structure (heterogeneity patterns and B.C.s) is identical to that of the regional model. The large scale domain extends up to the natural lateral edges of the Quaternary Terrace. Thus, zones of constant transmissivity were those of Abarca et al. (2006) which were based on a detailed sequential stratigraphy analysis by Gámez et al. (2009). A Neumann type boundary condition was prescribed identical to the regional model for these edges to account for the inflow from lateral creeks (marked in Figure 1C already and they will be added to Figure 1A). This lateral inflow is probably non uniform in space, but was treated as uniform both for simplicity and because alluvial fans probably distribute this inflow spatially. The Northern and Southern edges of the large scale domain were defined according to two batteries of piezometers located perpendicularly to the regional flow. We prescribe the piezometric head in those boundaries using a Dirichlet type boundary condition.

(2) A multilayer local scale domain was adopted for the zone affected by the tacer experiment. Its triangular shape of the local scale domain is based on a particular transmissivity zone defined in the regional model. This facilitated calibration, discussed below. Nine layers were adopted to represent aquifer layering (see section 3, below).

These two models are totally coupled. That is, both are solved together in every model

run. That is, the division is made for practical reasons (it would not make sense to extend layering to the full domain, because it would neither be possible to calibrate, nor it would affect the results. It would have been possible to solve first the large scale model, second, extract heads at the edges of the local scale model and, third, treat these edges as Dirichlet boundaries using those heads. Results would have been identical and we would have saved some CPU time. However, we did not do it because the saving is not dramatic and, with our tools, it would have been tedious (we would have had to transfer head at every node and time step).

The calibration and modeling strategy consisted of three steps. First, starting from the parameterization of the regional model by Abarca et al. (2006), we used updated meteorological and piezometric head data. Meteorological data was used to compute recharge and lateral inflows using the same procedures as for the regional model. The large scale model was "recalibrated" using the newly collected heads and the transmissivity values of the regional model as prior estimates. As it turned out, changes from the values of the regional model were minimal. Second, we calibrated the porosity and hydraulic transmissivity of the local scale domain, and the preferential flow through the reactive barrier using the piezometric head and amino-G acid concentrations obtained from the tracer test. We performed the calibration for the homogeneous and heterogeneous scenarios. Third, we validated the model by reproducing observed values of TCA and EC (see Section 4, below).

3. Heterogeneity structure.

As mentioned above, the area around the recharge basin was simulated as multilayer to be able to reproduce measurements and to analyze the effect of layering. The 14m-thick aquifer was divided into seven 2m-thick layers in the local scale domain. These 2m-thick layer emulate the material differences of the alluvial deposits. There are two additional 0.3m layers inside this domain representing the reactive barrier of the infiltration basin. The number of layers was chosen to obtain a sufficient precision in the vertical discretization while maintaining the numerical burden below a reasonable level.

Increasing the number of layers would have allowed us a finer discretization and better numerical accuracy but would have also increased computational and processing times. Each of the layers was assigned flow and transport parameters representing the properties of the aquifer materials, as derived from the cores of the boreholes drilled in the area. Each layer was homogeneous in the horizontal direction. It is clear that this is a simplification. Horizontal variation is expected both for the horizontal permeability and, especially, the vertical permeability (fine sediments layers, which control vertical permeability, are probably not continuous in reality). We simulated them as continuous both for simplicity and for robustness. It is clear that the calibrated permeabilities represent effective values, but are probably very sensitive to the location of measurement points. This is why we felt we had to perform validation runs (see Section 4, below). From the numerical point of view, layering is simulated using a quasi-3D approach, where horizontal connectivity is simulated via (sub)horizontal triangular elements. These elements are linked by 1D elements that reproduce the vertical connection between the layers. The approach is similar to cell centered finite differences or finite volumes. It is also similar to prismatic finite elements (actually identical if two integration points are used along the vertical direction). However, we find it more practical from a parametrization point of view in that our approach facilitates parametrizing separately the horizontal conductivity (controlled by sand layers) and the vertical conductivity (controlled by the fine sediments layers). The hydraulic conductivity of the 1D elements located at the edges of the local scale domain is very high to avoid a barrier effect where the monolayer and multilayer domains merge. The parameters of the rest of 1D elements are such the vertical water and solute fluxes are well represented.

4. The description and motivation of the used "tracers" (amino-G, TCE, EC)

Because of all the simplifications described above, the final model might have been an artifact. Therefore, we felt that it was necessary to test its validity. To this end, we simulated the evolution of both recharge and aquifer tracers to simulate flow and transport both during periods of time much longer than those used for calibration and comprising intervals of both artificial recharge and non-recharge. The selection of tracers was based on an "opportunistic" basis: 1) Amino-G was selected as tracer because it is easy to analyze with high accuracy and precision using fluorometers (we also injected a metal complex, but did not use it for calibration). 2) TCA was selected as a tracer because it was already present in the aquifer but not in the recharge water. Therefore, it complements the data of the artificially added amino-G acid. Furthermore, it provides information about the rate at which aquifer water returns to the space occupied by recharge water once recharge stops. 3) EC data was also used to validate the model. EC is highly variable in time both in the river (i.e., recharge water) and the aquifer, because high salinity comes from the salt mines located in the Llobregat River Basin far upstream from our site. The large amount of EC data (from 2012 to 2014) available in most of the monitoring points allowed us to evaluate the model under flow conditions different from those prevailing during the tracer experiment. We stress that no calibration was made using TCA or EC measurements. Validation simply consisted of changing the modeled time interval, and changing initial and boundary concentrations, as well as concentration of recharge inflow (zero for TCA, and continuously recorded at the infiltration basin for EC). As shown in the paper, results were very good, far better than we had anticipated, although actual recovery of aquifer water when recharge stops was a bit faster than modeled, which leads as to conclude that a MRMT model might have done a better job at reproducing the effect of unmodelled heterogeneity, albeit at the cost of added complexity.

References

- Abarca, E., Vázquez-Suñé, E., Carrera, J., Capino, B., Gámez, D., and Batlle, F.: Optimal design of measures to correct seawater intrusion, Water Resources Research, 42, n/a–n/a, doi:10.1029/2005WR004524, w09415, 2006.

- Gámez, D., Simó, J. A., Lobo, F. J., Barnolas, A., Carrera, J., & Vázquez-Suñé, E. (2009). Onshore–offshore correlation of the Llobregat deltaic system, Spain: Development of deltaic geometries under different relative sea-level and growth fault influences.

Sedimentary Geology, 217(1), 65-84.

- Hidalgo, J. J., a. S. L.,Medina, A., and Carrera, J.: A Newton-Raphson based code for seawater intrusion modelling and parameter estimation, in: Groundwater And Saline Intrusion Selected Papers from the 18th SaltWater Intrusion Meeting, 18th SWIM, IGME, Madrid, S., 15, pp. 111–120, IGME, doi:84-7840-588-7, 2004.

- Medina, A. and Carrera, J.: Coupled estimation of flow and solute transport parameters, Water Resources Research, 32, 3063–3076, doi:10.1029/96WR00754, 1996.

- Medina, A. and Carrera, J.: Geostatistical inversion of coupled problems: dealing with computational burden and different types of data, Journal of Hydrology, 281, 251 – 264, doi:10.1016/S0022-1694(03)00190-2, 2003.

- Walther, M., 2016, Comments to "Tracer test Modeling for Local scale characterization of an artificial recharge site", HESS Discussions, http://editor.copernicus.org/index.php/hess-2016-197-RC2.pdf?_mdl=msover_md&_jrl=13&_lcm=oc108lcm109w&_acm=get_comm_file&_ms=51089&c=108491&salt=1369714

---

## Referee Comment (RC3) · Anonymous Referee #3 · 16 Jul 2016

The authors investigated a MAR system near Barcelona, Spain using tracer methods combined with numerical analyses. Here they highlighted the importance of heterogeneous structures on the estimation of travel times of waters below an infiltration basin which is of importance when it comes to assessment of removal rates of unwanted substances. The manuscript is clearly written and introduction fits to the content. The methods used and the results obtained are of high scientific relevance. Beside this, the methods and analyses used seem appropriate regarding the objectives focused on. Therefore, I suggest to accept the manuscript after some minor revisions by authors. Please see the following comments:

The main drawback of the manuscript in its current state is the simplified explanation

of the model set-up used here. Although the methods seem scientifically correct, additional information should be provided to the reader to support a better understanding of the content and to ensure reproducibility of the numerical results. This holds true for both the input parameters used here and the numerical procedure. Regarding the first, for example, dispersivities in horizontal and transverse directions cannot be found in the manuscript. However, often dispersivities are used to include the effect of mixing and spreading if spatial variability of the hydraulic parameters is not included directly in the model. This may lead to larger dispersivity values used in homogenous models as compared to heterogeneous models for the same conditions. Are dispersivities for both models the same?

Regarding the numerical procedure some additional information would be helpful, e.g. the software used and how the inner region was selected in its extents (why basin so close to local domain boundary). Beside this, for me it is not clear if and in case of any which retention model was used for the unsaturated zone modeling?

Next to the mentioned issues, the link between lithology data from the site and the layering used in the models should be better explained (maybe using some lithology profiles as shown in Fig. 1). In page 6, Line 12 and page 8 line 9-10 preferential flow paths were mentioned which indicate non-continuous layering. Are there additional bore profiles, than shown in Fig. 1 supporting the assumption that layers are consistent at the site? Is there any change of thickness in the layers with space in the model? Please explain this further as the fact that every layer is exact 2 m in thickness everywhere sounds a little bit subjective.

Specific comments: Page 1, Line 5: "broad" seems rather undefined. Please better define.

Page 1, Lines 7-9: Please check the order of these two sentences. The heterogeneous model is mentioned after the 9 layers which are part of this model.

Page 2, Line 8: "the wells" – At this point nobody knows about the observation wells of

this study.

Page 2 Line 9-10: What is the difference between point (1) and (2): In both cases the concentration of substances are observed and conclusions are made.

Page 2, Line 14: Flux distribution may also be influenced by temporal flow field changes and different sources and sinks of water balance within a region.

Page 2, Line 17-19: Vertical distribution of hydraulic properties can be gained among others using Direct-Push techniques (e.g. Dietrich et al. 2008, Butler et al. 2002).

Figure 1: The gamma-ray profile and the lithology should be shown in Fig 1B.

Page 5, Line 22: Is the large domain the regional model sometimes mentioned? Please clarify procedure here.

Page 5, Line 30: Which kind of local tests do you mean?

Page 6, Line 10: How this function look like? Is there an areal distribution of the input concentration?

Page 6, Line 14: Why both layers, layer 7 and 9 were used distributing the time-dependent inflow data?

Fig. 3: Different scales make the concentrations hard to compare with each other. Maybe these could be transformed into equal scales (maybe using logarithmic scales?)?

Page 11, Line 7: typo "iwas"

Page 11, Line 25 Is TCA possibly subject to reaction (conservative modeling used here)?

Page 11, Line 33-34: Maybe velocity field without artificial recharge is not very sensitive to local hydraulic conductivities, but maybe other processes were missed in the model as it was built and calibarted to reproduce the vertical infiltration processes.

Fig. 5 and 6 and statements on page 11 line 32ff: What is the screened portion of the Px named observation wells? Does the screened interval match with the lithologic layering used in the model (for all observation points, also P8.x wells) and the respective concentrations simulated in the layers?

――――――――――――――――

---

## Author Comment (AC3) · 25 Jul 2016

We thank the reviewer for his/her constructive comments and the help to improve the manuscript. As we did with the other reviewers comments and in the spirit of the HESS discussion, we debate below the points that might need further explanations. Detailed responses to all comments from the three reviewers will be composed and sent with the revised manuscript.

Some of the most important issues highlighted by reviewer 3 , such as the modeling tool strategy, calibration strategy, the way the local and the large scale models were coupled, and the reason to divide the aquifer into seven 2m-thick layers, were also raised by the reviewer Prof. Marc Walther. Answers to those issues are discussed by

Valhondo et al (2016) and will not be discussed here. Other issues highlighted are addressed below.

1- Amino G acid input time function and infiltration basin zones:

The amino G acid input was modeled as a time dependent source term that distributes the total used mass (8 kg) over a 15 minutes period. In reality, the tracer was poured at the entrance of the infiltration. Presumably, it travel on the basin so that it was not homogeneously distributed in the whole basin water volume. This together with the potential degradation of amino-G acid resulted in a higher concentration of amino-G acid close to the infiltration basin entrance than in the rest of the basin. Actually, the amino-G acid concentration measured in P8.3, placed close to the infiltration basin entrance, was 2.75 times higher than the expected concentration for an homogeneous dilution. To emulate this process we divided the infiltration basin in nine zones and applied a weighting factor to the amino-G acid time function for each zone. The factor is greater than one in the zones close to the entrance of the infiltration basin and lower than one in the rest of the zones. The mass balance was taken into account to ensure that the total amino-G acid mass introduced was precise (6683 mg in Het-1, 6853 mg in Het-2, and 7885 mg in Hom) . In the revised manuscript we will add to table 1 the information about the amino-G acid mass for each outcome (Het-1, Het-2, and Hom).

2- Preferential flow paths:

The assumption that flow through the reactive barrier occurs through preferential flow paths is suggested by the redox indicators species measured in the infiltration basin, the suction cups, and the monitoring point P8.3. It is discusses by Valhondo et al., (2014 and 2015). Indications of preferential flow paths include: (1) very fast arrival of the tracer (it would have taken several days if flow was uniform), and (2) presence in P8.3 of oxidizing species (e.g. NO3-) and disappearance of reducing species that had been sampled in the unsaturated zone (e.g. Fe+2). We considered the heterogeneity of the reactive barrier and the high permeability of the aquifer (clean sand and gravel)

as the main causes favoring the flow of the recharge water along fingers and traveling at a velocity similar to the hydraulic conductivity (Hill and Parlange, 1972; Selker et al., 1996; Cueto-Felgueroso and Juanes., 2008). To emulate the fast infiltration of recharge water through these preferential flow paths we distributed the recharge water volume entering into the infiltration basin between layer-9 (representing the surface of the reactive barrier) and the surface area of the infiltration basin projected on layer-7 (representing the aquifer). As it turned out, best fit was obtained by distributing inflow 60% and 40% between layer-9 and Layer-7.

3- TCA reactivity:

TCA is very persistent in the environment with reported half-life times of 136-720 days in groundwater (Zhao et al., 2015). TCA can be degraded by biotransformation or abiotically. TCA biotransformation occurs mainly through the process of reductive dechlorination for which anaerobic conditions are desired and DOC is used as substrate . The natural groundwater of the aquifer is mainly aerobic, with average $O_2$ concentration of 3mg/l and average DOC concentration of 2.5mg/l. Therefore, we expected little biodegradation and half life time higher than the model residence time. The abiotic TCA transformation into DCE and acetate by common hydrolysis in water happens as a first-order kinetic law with a half-life time of 2.9 years at 15 °C (Lookman et al., 2004), which is much longer than the residencen time in the local domain. Therefore, TCA can be considered as a conservative tracer in the time scale of the model.

References

-Cueto-Felgueroso, L., Juanes, R.: Nonlocal interface dynamics and pattern formation in gravity-driven unsaturated flow through porous media. Phys. Rev. Lett. 101, 244504. doi:10.1103/PhysRevLett.101.244504, 2008

-Hill, D.E., Parlange, J.Y., 1972. Wetting front instability in layered soils. Soil Sci. Soc. Am. J. 36, 697–702, doi: 10.2136/sssaj1972.03615995003600050010x, 1972

- Lookman, R., Bastiaens, L., Borremans, B., Maesen, M., Gemoets, J., and Diels, L.: Batch-test study on the dechlorination of 1,1,1-trichloroethane in contaminated aquifer material by zero-valent iron, Journal of Contaminant Hydrology, 133–144, doi: 10.1016/j.jconhyd.2004.02.007, 2004.

-Selker, J.S., Steenhuis, T.S., Parlange, J.-Y.: An engineering approach to fingered vadose pollutant transport. Geoderma 70 (2–4), 197–206, doi: 10.1016/0016-7061(95)00085-2, 1996.

- Valhondo, C., Carrera, J., Ayora, C., Barbieri, M., Nödler, K., Licha, T., and Huerta, M.: Behavior of nine selected emerging trace organic contaminants in an artificial recharge system supplemented with a reactive barrier, Environmental Science and Pollution Research, 1–12, doi:10.1007/s11356-014-2834-7, 2014.

- Valhondo, C., Carrera, J., Ayora, C., Tubau, I., Martinez-Landa, L., Nödler, K., and Licha, T.: Characterizing redox conditions and monitoring attenuation of selected pharmaceuticals during artificial recharge through a reactive layer, Science of The Total Environment, 512–513, 10 240 – 250, doi:10.1016/j.scitotenv.2015.01.030, 2015.

-Valhondo et al.: Reply to comments by Prof. Marc Walther, Interactive discussion HESSD, Url: http://editor.copernicus.org/index.php/hess-2016-197-AC2.pdf?_mdl=msover_md&_jrl=13&_lcm=oc108lcm109w&_acm=get_comm_file&_ms=51089&c=108879&salt=1312152
2016

- Walther, M.: Comments on "Tracer test modeling for local scale residence time distribution characterization in an artificial recharge site", Interactive discussion HESSD, Url: http://editor.copernicus.org/index.php/hess-2016-197-RC2.pdf?_mdl=msover_md&_jrl=13&_lcm=oc108lcm109w&_acm=get_comm_file&_ms=51089&c=108491&salt=1257066
2016.

- Zhao, S., Chang, D., Jianzhong, H.: Detoxification of 1,1,2-Trichloroethane to Ethene by Desulfitobacterium and Identification of Its Functional Reductase Gene, PLoS ONE,

1-13, doi: 10.1371/journal.pone.0119507, 2015.

---

## Author Response (AR1)

**R1.1** Valhondo et al. (2016) is an important paper that exams near-field flow under managed aquifer recharge (MAR) spreading ponds. It combines both geochemical tracer techniques and numerical modeling of flow and transport. The combination of these separate approaches reveals the complexity of flow beneath spreading ponds. I suspect that this is largely due to the local hydrogeology. Spreading ponds are often located in alluvial settings such as the one examined near Barcelona, Spain by the authors.
A unique contribution of this paper is the careful consideration given to the vertical structure of the aquifer below the spreading pond. The authors created a finer scale grid with nine layers that was imbedded into a regional numerical model of flow by Abarca et al. (2006). Three different fine scale models were tested: "Hom"—containing homogeneous Kh and Kz; "Het-1" —containing different Kh for each of the nine layers while maintaining the same Kz for each layer; and "Het-2"—containing different Kh and Kz for each of the nine layers (see Table 1 of the paper). Field measurements of head and geochemistry were used to validate the fine grid model. I particularly appreciated the use of TCA (1,1,2-trichloroethane) as way to quantify the advection of regional groundwater into the study area.
The most significant contribution of Valhondo et al. (2016) is their characterization of preferential flow in the heterogeneous aquifer found in their study area. Unfortunately, they did not discuss the work of Thompson et al. (1999) who produced one of the original tracer data numerical flow models to interpret the complexity of flow and transport near MAR sites. They used a different approach but reached a similar conclusion. As mentioned above, MAR sites are more often than not located above heterogeneous aquifers, so the authors' findings should be applicable to other settings. As Fox et al. (2007) demonstrated many years ago at the 6th International Symposium on Managed Aquifer Recharge (ISMAR6), the placement of monitoring wells for management purposes must account for preferential flow. Without using complex numeral model such as those employed by Thompson et al. (1999), Fox et al. (2007) and Valhondo et al. (2016) or detailed deliberate (added) tracer experiments (e.g., Clark et al., 2014; Becker et al. 2014), it is hard to demonstrate the residence time distribution and hydraulic connection between the recharge area and monitoring well. Therefore documentation of water quality changes is uncertain and must be recognized.

**R1.1**- We thank the reviewer for his kind assessment of our work and for his added insights into the role of heterogeneity. We agree. Heterogeneity is an essential feature of most aquifers. In fact, layering and/or channels should be expected in most sedimentary aquifers. This can be well reproduced using tools such as transition probability models (Carle and Fogg, 1997), which was beautifully demonstrated in the Orange County case  (Thompson et al.,1999). Heterogeneity causes uncertainty (Park et al., 2006) and promotes a broad range of residence time distributions (Thompson et al., 1999). Moreover, this broad range, together with flux fluctuations (driven by variations in natural and artificial recharge) favors mixing of different waters. We contend that this mixing contributes to water quality improvement. In the revised version, we will expand the discussion on the effects of heterogeneity to acknowledge the reviewer´s comments.

I also found a few typos in the

paper that you may want to fix:

**R1.2** P. 2, line 23: Becker et al. (2015) completed their work at the San Gabriel Spreading Grounds Test Basin in Los Angeles County, CA not in Orange County, CA as stated in the text. The paper was published in 2014 not 2015.

**R1.2**- It has been corrected in the revised document.

**R1.3** P. 7, line 1 (below figure): Typo. "Figure2" should be "Figure 2"

**R1.3**- It has been corrected.

**R1.4** P. 11, line 7: Typo. "iwas" should be "was"

**R1.4**- It has been corrected.

**R1.5** P. 15, line 16: Typo. "10B" should be "10B" (10 needs to be superscripted).

**R1.5**- We rectified it.

**R1.6** P. 15, line 17: Typo. "2015" should be "2014".

**R1.6**- It has been rectified

Responses to Comments by reviewer 2 .

**R2.1** Comments on "Tracer test modeling for local scale residence time distribution characterization in an artificial recharge site" submitted to HESS by Valhondo et al. The manuscript features investigations on a tracer test in an artificial recharge site by utilizing a simulation approach. Different implementations of the study area are realized with homogeneous and heterogeneous hydrogeological setups. The motivation is to understand the relevance to implement different (heterogeneous) layers in order to represent correct flow and mixing behavior of juvenile and upstream recharge waters to evaluate the performance of AR for contaminant removal. A key factor for this is the residence time distribution which was measured in a field campaign by break through curves of a tracer test. The submitted manuscript is structured in a logical way, giving a comprehensive introduction and motivation, before presenting materials and methods, showing and discussing results, and finally, drawing some short conclusions. The work is of high quality, written in a clear and understandable way, while some figures and tables support the text, and relevant references are cited appropriately. The authors manage to resemble the measurements with the model software remarkably well, albeit the relatively complex study area. By comparing results from homogeneous and heterogeneous setups, conclusions clearly show deficits of a homogeneous setup. Yet, still some questions remain open for further investigation (e.g. uniqueness of two calibrated heterogeneous setups). Already with these two aspects, I think that the manuscript is of high relevance for current research. Nevertheless, the manuscript also can be improved, especially in the following major points: a) The description and motivation of the used

"tracers" (amino-G, TCE, EC) should be given in a clearer way for the reader to understand which is used for what purpose. This may be done by giving a short overview in the beginning of the methods section. b) The information given on the modeling tool are too scarce. A very short description on the type of the tool, and its features should help to understand the decision to use this tool. Furthermore, the description on the modeling strategy, and the implementation of the model (model setup) could need some more structuring (clear description of all boundary conditions, section 2.3, and full list of calibrated parameters). Also, information on the calibration strategy should be provided. c) Finally, I would like to encourage the authors to state a more profound argumentation why they set up the heterogeneous models in the way they did. For example, why were 9 layers chosen and not 5, 15, or 40? I think that this can aid to support their conclusions, ie. to highlight relevance of a layered model structure in the study area. With these general comments, I enclosed a list of more specific ones. If the authors decide to reply to these comments, I would be available for a second revision, if the authors think that my comments gave a valuable input. I think, that this manuscript is already of high quality and tackles an important topic. I, therefore, suggest to consider the submission for publication after revision.

**R2.1**- We thank Prof. Marc Walther both for his kind assessment of our work and for the time he has devoted to improving the paper, as can be derived from the length of his comments .

We extended  the model description in order to clarify the roles of each tracer, and to provide further information regarding the modeling tool and strategy.

**R2.2** Abstract: Page 2, Line 8: You say that you used the numerical model to "...extend characterization to other flow conditions...", but I could not find any scenario simulations that are different from the observation times in the manuscript. Could you please rephrase this?

**R2.2**-There are two sides to this comment. First, in general, you do models to assess how the aquifer system behaves during calibration and, usually, under different flow conditions. But, second, in our case, local scale flow and transport calibration extended for two months, whereas model testing (validation) extended for two years. Flow conditions during this time were highly variable and different from those during calibration.

Local domain parameters were calibrated using head and amino-G acid data from the tracer test which extended for over two months (July-September 2012). In this period the recharge system was working and it stopped at the beginning of August. The estimated parameters were validated by reproducing more than two years of  EC measurement  and eight months of TCA concentrations in samples collected during 2011 with the system operational and with the system stopped. In both cases the flow conditions changed (system operational and not, and regional levels varied compared to the tracer test period) and therefore we test the validity of the model under flow conditions diverse from those prevailing during the tracer test.

**R2.3** Page 3, Line 24: You write "Characterizing heterogeneity in such systems at the recharge basin scale is required for proper representation of RTDs. But it is hard because the head differences are small and detailed hydraulic testing difficult to perform." - While the mentioned issues are problematic indeed (ie. difficulties with small head differences, and to get detailed information of hydraulic properties), I would also add, that measuring hydraulic

properties and the distribution of the heterogeneity (micro and macro scale) is unfeasible, as you would never be able to cover the whole basin in the full resolution.

R2.3- Indeed, we agree. We have added a sentence to emphasize the point.

R2.4 P4, L7: "The objective of this paper is to describe the tracer test and its interpretation using both heterogeneous and homogeneous models to assess the need for model complexity." - The objective of paper you describe here is somewhat different from the one you give in the title: here, it sounds as if you wanted to evaluate the necessity of complexity in a numerical model, while the title leads to the intention that you want to model RTD in an AR site. Can you please synchronize these locations (ie. add the missing part in the text or rephrase the title)?

R2.4-The RTD is a result of heterogeneity. As such, proper reproduction of RTDs requires acknowledging in the model the most salient features of the K field, the goal is to reproduce RTDs because we expect them to inform about both mixing, which controls fast reactions, and spreading, which controls kinetics. Still, the reviewer is right. The link is not immediate. In order to synchronize, we have revised the title and extended the text about the objective.

R2.5 Figure 1: * (C) - Can you please add the location of the cross-section in (A) and/or (B)? - I do not understand what the blue arrows indicate - can you please explain these? - What are "thick" and "fine" sediments? - Can you please add a legend to the stratigraphic log (or add the description of the layers right next to it)?

R2.5- The cross-section location has been added in A in the revised manuscript. The big blue arrows indicate the regional flow direction, the thin blue arrows are used just to link text with the figure when the figure did not allow to include the text. Thick and fine sediments refer to sediment size of the alluvial aquifer. The stratigraphic log is a general sequence observed in one monitoring point to illustrate layering. It was not meant to represent the defined model layering but just to display the typical alluvial sequence.

R2.6 P6, L2: You write that "The Llobregat River was disconnected from the aquifer...", but the arrow in fig 1B shows a groundwater flow direction parallel to the river; can you please explain how you came to the conclusion of a disconnected river?

R2.6- Both the Llobregat River and the aquifer flow towards the sea, but the head of the aquifer was between six and eight meters below the surface and far below from the River bed. We have also profited from previous work on the interaction between the Llobregat River and its alluvial aquifer. In essence, high recharge from the river occurs during flood events, which remove the low-K clogging layer at the river bed. But the high load of fine sediments in suspension in river water ensures that the river bed gets clogged again in a few weeks (Vazquez-Suñé et al., 2006). As a result, infiltration is small most of the time. We cite this paper in the revised version, but we feel that the full explanation would be distracting. Also, we move this statement to section 2.3.1.

R2.7 P6, L3: A question out of curiosity on the operation of the AR: what is the reason to divert the river water firstly to a "settlement basin" - is this due to high turbidity of the river water? Can you please provide an estimate of the volume of the basins (you only give an area here)?

**R2.7**- Yes indeed, to reduce clogging at the infiltration basin, the water remains in the settlement basin approximately 3 days where most of the suspended solid is expected to settle. The levels in the settlement basin were about 2-2.5 m whereas in the infiltration basin the levels hardly ever exceeded the 50-70 cm, but this is not easy to assure due to the irregularities in the basin surfaces.

**R2.8** P6, L5: How was the flow rate measured?

**R2.8**- It was measured using an area velocity flow meter (Teledyne Isco Inc, Lincoln, Nebraska, United States) located in the pipe connecting the settlement and the infiltration basins.

**R2.9** P7, L6: By writing "depths of 7 m" etc, you mean "meter below ground surface" or "meter below groundwater level"?

**R2.9**- We mean meter below ground surface. It has been added to the revised manuscript.

**R2.10** P7, L7: Can you please tell in which layers or depth the piezometers P2, 5, 9, and 10 are filtered?

**R2.10**- These four piezometers are fully screened. During the test, the head gradient is downward. Therefore, recorded concentrations should be those of the topmost high conductivity layer at each piezometer. We have added the information in the revised manuscript.

**R2.11** P7, L12: Can you please add the location of the addition CTD-diver to the map (fig 1)?

**R2.11**- The additional CTD-diver was located beside piezometers P8.3, P8.2, and P8.1. This has been added in the revised manuscript , section 2.1 "Site description and instrumentation".

**R2.12** I like the detailed description of the measuring campaign; this provides a solid information for the interpretation of the results!

**R2.12**-Thank you so much!

**R2.13** Section 2.3.1 needs to be clearer. In the first paragraph, it is not obvious, which combination of boundary conditions you used to acquire information for the local domain BCs, and which type of BCs you implemented. Eg: P8, L23: You describe that you added inflow from "eastern and western local creeks". - I cannot find them in figure 1; can you please consider adding them in figure 1 for a better overview? - I assume, you used a Neumann-type boundary condition for implementing the creeks (as you write "inflow"). What type of BC did you use in the regional model? Did you also include the abstraction of the drinking water wells in the regional model to get the values for the creeks' inflow? - Why did you model the eastern part from the Llobregat river? Is this part of the domain relevant for the western model domain?

**R2.13**- The regional model boundaries were the natural edges of the Llobregat delta main aquifer. Our model, which is limited to a small portion of the Llobregat River alluvial aquifer, increases the detail at the local scale (around the infiltration system).

The large scale domain extends up to the natural lateral edges of the Quaternary Terrace. Thus, zones of constant transmissivity were those of Abarca et al. (2006) which were based on

a detailed sequential stratigraphic analysis by Gámez et al. (2009).  A  Neumann type boundary condition was prescribed identical to the regional model for these edges to account for the inflow from lateral creeks (marked in Figure 1C already and they will be added to Figure 1A). This lateral inflow is probably non uniform in space, but was treated as uniform both for simplicity and because alluvial fans probably distribute this inflow spatially.  The Northern and Southern edges of the large scale domain were defined according to two batteries of piezometers located perpendicularly to the regional flow. We prescribe the piezometric head in those boundaries using a Dirichlet type boundary condition. The local scale domain was adopted for the zone affected by the tracer experiment. The triangular shape of the local scale domain is based on a particular transmissivity zone defined in the regional model to facilitated calibration. The two domains, large scale and local, are totally coupled.  That is, both are solved together in every model run. The division is made for practical reasons (it would not make sense to extend layering to the full domain, because it would neither be possible to calibrate, nor it would affect the results. It would have been possible to solve first the large scale model, second, extract heads at the edges of the local scale model and, third, treat these edges as Dirichlet boundaries using those heads. Results would have been identical and we would have saved some CPU time. However, we did not do it because the saving is not dramatic and, with our tools, it would have been tedious (we would have had to transfer head at every node and time step).

We have included more information to clarify the two scale domains and the prescribed boundaries conditions.

**R2.14** Why did you choose a triangular local model area? Wouldn't an area that is limited on the lateral sides by flow paths that provide a no-flow boundary be a better solution?

**R2.14**- You are probably right (but bear in mind the flow paths are highly variable). The triangular local domain is based on a particular transmissivity zone defined in the regional model  in which the recharge system is contained. So the choice was make for modeling convenience.

**R2.15** P9, L4: Add comma in "(local domain ≈0.5 x 1.5 km2, yellow triangle in Figure 1 A),"

**R2.15**- A comma has been added.

**R2.16** Section 2.3.2 is a little confusing: I expected that you talk about the model setup of the local scale model (yellow triangle in fig 1), as you motivated the setup of this smaller model from an existing regional scale model in 2.3. Section 2.3.1 then gives BCs etc for the small scale model, as 2.3.3 give more details on it. In between, section 2.3.2 feels a little lost. Maybe, the information you provide there (hydraulic parameters, head data), could be given before 2.3.1? Besides this, the heading of 2.3.2 does only suspect info on the large scale model, while there is actually also some info on the local domain.

**R2.16**- We have changed the heading of the sections and modify some information to make these sections more understandable.

**R2.17** P9, L8: You say that there is a strong local heterogeneity (layering due to fluvial deposition). Can you please motivate your decision to use exactly 9 layers? How many

hydrogeological layers can you find in the domain? Do these 9 model layers represent major (important) hydrogeological layers at the site? What would happen, if you used 100 layers (ie. using more than one grid layer per hydrogeol. layer)? Can these 9 layers represent vertical distribution of a tracer within one hydrogeol. layer?

**R2.17**- The number of layers was chosen to obtain a sufficient precision in the vertical discretization while maintaining the numerical burden below a reasonable level.  Increasing the number of layers would have allowed us a finer discretization and better numerical accuracy but would have also increased computational and processing times. Each of the layers was assigned flow and transport parameters representing the properties of the aquifer materials, as derived from the cores of the boreholes drilled in the area. Each layer was homogeneous in the horizontal direction. It is clear that this is a simplification. Horizontal variation is expected both for the horizontal permeability and, especially, the vertical permeability (fine sediments layers, which control vertical permeability, are probably not continuous in reality). We simulated them as continuous both for simplicity and for robustness. The calibrated permeabilities represent effective values, but are probably very sensitive to the location of measurement points. Ultimately, this is why we felt we had to perform validation runs.

**R2.18** P9, L8: You write, that the nine layers "overlapped in the local domain". Do you mean vertically overlapping of the elements (ie. you get prism elements) or elements overlapping in the way of "being at the same location"?

**R2.19** P9, L9: You state, that you "linked" the layers "by one-dimensional elements". - What is the motivation to do so? Are the layers you introduced in the 3d part of the model domain disconnected initially? Do these linking elements exist for the communication of the layers? - Did you link at nodes or elements? - How many 1d elements did you use to link two layer's nodes/elements (only one line)?

**R2.18 and R2.19**- Actually both. From the numerical point of view, layering is simulated using a quasi-3D approach, where horizontal connectivity is simulated via (sub)horizontal triangular elements. These elements are linked by 1D elements that reproduce the vertical connection between the layers. The approach is similar to cell centered finite differences or finite volumes. It is also similar to prismatic finite elements (in fact, identical if two integration points are used along the vertical direction). However, we find it more practical from a parameterization point of view that our approach facilitates parameterizing separately the horizontal conductivity (controlled by sand layers) and the vertical conductivity (controlled by the fine sediments layers). The hydraulic conductivity of the 1D elements located at the edges of the local scale domain is very high to avoid a barrier effect where the monolayer and multilayer domains merge. The parameters of the rest of 1D elements are such the vertical water and solute fluxes are well represented.

**R2.20** P9, L10: You describe, that you introduced two top layers (8 & 9) for the reactive barrier. Assuming, that the preferential flow direction is vertical, and considering that you use linear elements for vertical exchange, why is it necessary to use a second layer for this feature - wouldn't one be enough? Besides that, can you say, that this top reactive barrier layer is having homogeneous properties in horizontal extent? Could it be clogged preferably at the

inflow zone due to stronger biological activity and availability of nutrients? (Later, you also distribute the tracer along the infiltration bed, based on a similar assumption.) What would be the influence of such reactive barrier heterogeneity on the distribution of the infiltrating water/tracer?

**R2.20**- There are two aspects in this comment. We used two layers because we wanted to refined the reactive barrier zone. It is clear that the layer is heterogeneous, but we do not know the heterogeneity. High frequency variability is probably best simulated using effective approaches, like simulating that a portion of the recharge enters directly into layer-7. This represents the effect of the fingering through the sandy unsaturated zone and high K flow paths through the barrier. The effect of low frequency variability is reproduced by discretizing the recharge basin into nine zones. One of the effects of the biological and non-biological clogging and heterogeneity of the barrier is the preferential flow through it.

**R2.21** P9, L15: I do not understand, why you used the TCA for validation - I thought that you wanted to use the amino-G tracer for this? How did you probe for and measure TCA? Some TCA information is partly given in 2.4, but at this section it is not clear to me, why TCA is used. Maybe, you can write a short overview-like, introductory information on the intended usage of TCA and amino-G in 2.3 (or even 2.2)?

**R2.21**- Because of all the simplifications made during modeling, the final model might contain artifacts. Therefore, we felt that it was necessary to test its validity. To this end, we simulated the evolution of both recharge and aquifer tracers to simulate flow and transport both during periods of time much longer than those used for calibration and comprising intervals of both artificial recharge and non-recharge. The selection of tracers was based on an "opportunistic" basis:

1) Amino-G was selected as tracer because it is easy to analyze with high accuracy and precision using fluorometers (we also injected a metal complex, but did not use it for calibration). Therefore, we used it for the controlled tracer test.
2) TCA was selected as a tracer because it was already present in the aquifer but not in the recharge water. Therefore, it complements the data of the artificially added amino-G acid. Furthermore, it provides information about the rate at which aquifer water returns to the space occupied by recharge water once recharge stops.
3) EC data was also used to validate the model. EC is highly variable in time both in the river (i.e., recharge water) and the aquifer, because high salinity comes from the salt mines located in the Llobregat River Basin far upstream from our site. The large amount of EC data (from 2012 to 2014) available in most of the monitoring points allowed us to evaluate the model under flow conditions different from those prevailing during the tracer experiment.

We stress that no calibration was made using TCA or EC measurements. Validation simply consisted of changing the modeled time interval, and changing initial and boundary concentrations, as well as concentration of recharge inflow (zero for TCA, and continuously recorded at the infiltration basin for EC). As shown in the paper, results were very good, far better than we had anticipated, although the actual recovery of aquifer concentrations when recharge stops was a bit faster than modeled, which leads to conclude that a MRMT model

might have done a better job at reproducing the effect of unmodelled heterogeneity, albeit at the cost of added complexity.

**R2.22** P9, L17: What is "Transdens"? A modeling tool? Can you please give a little more information on this?

**R2.22**- The modeling tool used for calibration is TRANSDENS (Hidalgo et al.,2004 ), which is a development over TRANSIN (Medina and Carrera, 1996; Medina and Carrera, 2003). Both codes have been developed by the Hydrogeology Group (UPC-CSIC). Both use the Finite Element Method to solve the equations governing coupled flow and transport through porous media. A strict Galerkin method is used for transport, which places strong constraints on the adopted dispersivity, but which was not an issue in this case because of the strong heterogeneity. The singularity of these codes lies on their versatility to accommodate geology, zonation, time dependence of aquifer parameters and model inputs, and especially inverse modeling. That is, they allow automatic calibration of aquifer parameters (transmissivity, storativity, recharge, boundary heads and flows, leakage, dispersivity, molecular diffusion, porosity, retardation, linear decay, boundary concentrations) using the methods described by Medina and Carrera (1996) and Medina and Carrera (2003).  A short explanation will be included in the revised document in the section 2.3.1 "Boundary conditions and model parameterization".

 **R2.23** P10, L9: "...whole basin water volume, as a consequence the maximum..." - I suggest to divide this sentence (and add a comma): "...whole basin water volume. As a consequence, the maximum..."

**R2.23**- Done. It has been corrected in the revised version of the manuscript.

**R2.24** P10, L6 - L15: The reason to divide the basin into nine zones was hard to understand in the first place, as you state some results where the reader expects the model setup (which it indeed still is). Yet, there are some questions: What is the "amino-G acid concentration function"? Is this the concentration you measured at P8.3 (probably not)? Or is it defined through the input of amino-G at point A? How did you measure this function? Also, it is unclear to me, how you defined the different zones i-ix and on what basis you set values for the distribution parameters within the zones? Can you plot or state the distribution parameters in a table?

**R2.24**- The amino G acid input was modeled as a time dependent source term  that distributes the total used  mass (8 kg) over a  15 minutes period.  In reality, the tracer was poured at the entrance of the infiltration. Presumably, it travels through the basin so that it was not homogeneously distributed in the whole basin water volume. This together with the potential degradation of amino-G acid resulted in a higher concentration of amino-G acid close to the infiltration basin entrance than in the rest of the basin.  Actually, the amino-G acid concentration measured in P8.3, placed close to the infiltration basin entrance, was 2.75 times higher than the expected concentration for an homogeneous dilution. To emulate this process we divided the infiltration basin in nine zones and applied a weighting factor to the amino-G acid time function for each zone. The factor is greater than one in the zones close to the entrance of the infiltration basin and lower than one in the rest of the zones. The mass balance

was taken into account to ensure that the total amino-G acid mass introduced was precise (6683 mg in Het-1, 6853 mg in Het-2, and 7885 mg in Hom) . In the revised manuscript we will add to table 1 the information about the amino-G acid mass for each outcome (Het-1, Het-2, and Hom).

**R2.25** P10, L16: I don't think it is important to state that "time discretization was quite irregular". It is common to have a variable time stepping in a numerical model. The relevant questions for time step sizes would be, how you defined the different time step sizes (usually based on the error of an iteration, but here probably defined through the measurement time steps?), and whether large time steps still yield valid Courant numbers (if applicable for your model).

**R2.25**-Indeed, it is common, but we think that it might not be obvious for readers who are not so familiar with the modeling processes. The maximum time step was three days, which leads to high Courant numbers in 1-D elements. But TRANSIN is quite tolerant for Courant number violations.

**R2.26** P10, L21: Ly = Layer? Why didn't you distribute the inflow also on Layer 8? Did you distribute the inflow equally or weighted?

**R2.26**-We used abbreviation "Ly" because it is the same nomenclature displayed in Figure 4. Part of our objective was to resolve the portion of the inflow that flowed through preferential flow paths through the reactive layer and unsaturated zone (emulated in the model with layers 8 and 9) so we estimate this preferential flow by distributing the inflow between layer 9 and layer 7 (belonging to the aquifer and below the reactive layer). Distributing the inflow partly in Ly-8 would have led to a much richer effective model. But we would have lost robustness because we are deriving the inflow distribution from calibration, not from theoretically based considerations.

**R2.27** P10, L22: "The standard deviation assigned to all concentration measurements at each observation point was 1% of the maximum concentration at each point, which was to ensure that a comparable weight was given to each point during calibration (maximum concentration varied 25 by ≈2 orders of magnitude)" - I think this is an unfortunate decision. With this assumption, if an observation point has a low concentration, it must show a much more accurate measurement than a high concentration point. Thus, the points would be biased and low concentration points would be forced to perform much better. Wouldn't it have been better to use the standard deviations from the calibration/regression line of the fluorescence meters?

**R2.27**- The reviewer is right, we did it for simplicity. It is much better to adopt a heteroscedastic distribution. That is,, for each measurement, $\sigma_i$=Max $(\alpha C_i, \sigma_{min})$.

**R2.28** P11, L1: "...two different convergence points of the calibration process..." - Where there more than those two convergence points? - You say, that you defined two heterogeneous property distributions from the calibration. Could it also be, that the conceptual model has a fault, if the calibration does not produce a unique result? Did you consider other setups?

**R2.28**-This is an excellent point. Identifiability and non-uniqueness are facilitated if the conceptual model is at fault. But the opposite is not true. A good conceptual model may suffer from identifiability or display several local minima during calibration, when data are insufficient. In such cases, proper evaluation of uncertainty is difficult. Linear analyses of uncertainty are overly optimistic and non-linear are very hard in complex models. So, we did not try. Still, to give an idea of the kind of uncertainty in the model, we felt it would be appropriate to provide at least two alternative heterogeneous models.

**R2.29** P11, 2.4 Validation: - At this point of the manuscript, I do not understand, how you use TCA and EC data for validation. Do you try to calibrate porosity, dispersivity, retention or decay coefficients? Why do you need two parameters for this? Maybe, you can give a short, comprised overview of what tracers you use for what purpose in the introduction? - Because of the first point here, it is also hard to understand the usage of the boundary conditions: + Why do you set TCA to the maximum value but EC to the mean? + Why do you use measured EC time series as the recharge BC? + What are the "lateral inflows"? Can you show that in fig 1? + Did you simulate decay of the TCA? + I am sorry, but I do not understand this sentence: "TCA and EC concentrations in the northern border of the local domain were prescribed based on the TCA and EC measured at P1 and adjusted for the travel time from the northern border to P1."

**R2.29**- TCA and EC were used just to verify that the parameters calibrated using the amino-G data were feasible to reproduce other observations (see R2.21).

For EC the amount of data was much larger than for TCA, but again only for  monitoring points in the local domain. Therefore we had a quite detailed description of the EC evolution. We decided to use its evolution measured at P1 as the evolution that would be present in the aquifer. We prescribed that evolution on the northern edge of the local domain but taking into account the time that it would take the water to flow from the edge to the monitoring point P1. Lateral inflow referred to the water coming from the lateral creeks for which we fixed an average value of the measured EC.

TCA is present in the aquifer due to a historical contamination, but we do not know where the contamination comes from other than from north of the infiltration system. Since TCA concentration decreases over time even in monitoring point P1 and data from points upstream were not available we decided to prescribe the observed concentration in monitoring point P1 on the edge of the local domain in the same way that has been explained for EC. Lateral inflow concentration was fixed using the maximum observed value for TCA to account for the contamination source .

The procedure was identical for both, TCA and EC, but for TCA the number of observation was much smaller than for EC.

**R2.30** Figure 2: - Time unit of head series is probably not days (looks like months with years as axis).

**R2.30**-Indeed, it has been corrected in the revised version.

**R2.31** P13, L2: "...surface (Fig. 2). Figure2 displays..." - Please add a space in Figure2. - I think, you should also decide on using either "Fig." or "Figure".

**R2.31**- The space have been added. Figures have been referred in the text following the journal rules.

**R2.32** P13, L2: "...piezometers located around local domain." - I think, it was better to write something like "...piezometers located around the local domain.", or even "...piezometers located within the local domain."

**R2.32**- Changed. It has been corrected in the revised version.

**R2.33** P13, L3: "The fit was good,..." - Can you please give a quantitative measure (regression coefficient, RMSE, Nash-Sutcliff or similar)?

**R2.33**- Sure, it has been added in the revised version.

**R2.34** P13, L4: "The fit was good, which suggests that the size of the multilayer local domain was sufficient to reproduce head variations at the monitoring piezometers close to the basin where the gradient is mainly vertical (≈ 10%)." - Before, you said, that you produced three model variations; which one is used here to show fig 2? I would also argue, that if you have more than one unique solution (if the amount of measured data is enough), a good fit does not necessarily mean a sufficient model setup. - A gradient of 10% is a large value (10 cm head difference in 1 m, understandably for the AR), but not "mainly vertical". I suggest to rephrase this sentence to highlight the influence of the AR on the head distribution. - After reading explanations on fig 3, I understand, that you are referring to the vertical gradient itself. Please clarify this.

**R2.34**- It has been clarified in the revised manuscript.

**R2.35** P14, L2: "Conservative transport parameters were estimated for the nine layers of the local and basin domains using the tracer test data." - I would say, that this sentence should be given somewhere in 2.2 or 2.3, as it is more related to methodology than results. - Which "tracer test data" do you mean - TCA, EC or amino-G? - You never give a full list of transport parameters, please add this.

**R2.35**- We modified the methods sections in the revised manuscript to make it more explicit. Table 1 provides the values of the estimated parameters, other transport parameters such as horizontal and vertical diffusion coefficients remain equal for the three models.

**R2.36** Table 1: - What is the thickness of the layers? Can you add this to the table? Then, you could also easily compare and evaluate the results to the hydrogeological and gamma logs you give before. - Layer 5 shows a hydraulic conductivity of ~1000 m/d, which is ~1e-2 m/s, a relatively high value - can you justify this (and the other values) from the hydrogeology? I cannot evaluate it, as you do not give exact information on the sediments (apart from being fluvial). - Values for the reactive barrier seem relatively high, while I would expect them to be lower due to the finer material, clogging etc. Can you please explain this? - Also, in the reactive barrier layers, vertical conductivity is higher than horizontal - why? - Why are vertical porosity

values so extremely low? Can you give an explanation within a physical meaning for such small values? - Do I understand it correctly, Layers 9 and 8 are only present at the infiltration basin? If yes, what happens to the vertical elements of Layer 7 where there is no layer above? - What is the meaning of the RMSWE values for hydraulic conductivity? I know, how to get RMS(W)E for hydraulic head or concentrations (or whatever primary variable you calculate) but not for a model parameter, if you do not have otherwise measured reference values. Can you please explain this, together with how you weighted the values? - Where is the "input mass in outcomes Het-1, Het-2, and Hom, after calibration"?

**R2.36**-The thickness of the layers is 2m for layers 1, through 7, and 0.3m for layers 8 and 9 that emulate the reactive barrier and only covered the infiltration basin, as was mentioned in section 2.3.1.

The gamma ray profile was included just to show the clay level, no further comparison with the sediment properties are assumed. The layers defined in the model meant to reproduce the characteristic vertical heterogeneity of the alluvial sediments, but they do not correlate with real geological units. It is a simplification and the reason why the model needs to be validated.

The high hydraulic conductivity of this area has been largely mentioned in previous studies. The sediments are mainly coarse gravels and sands more or less sorted. In the heterogeneous models the initial values of transmissivity were obtained from a pumping test performed in the area in 2010. The obtained value was divided between the seven layers and afterward we calibrated the value for each layer. The results of both heterogeneous model yields to total transmissivity values comparable to the one obtained in the pumping test. In fact the total transmissivity of the three models are similar and consistent with the pumping test value.

The porosity of the reactive barrier was fixed at 0.5 for the three model. It is true that one might expect a lower porosity due to clogging but when the barrier was constructed the aquifer material was mixed with the vegetable compost generating a highly porous and little compacted material. The high porosity fixed for the barrier also leads to a higher residence time of the recharge water in these layers, which was the objective.

The reactive barrier is in the unsaturated zone and the infiltration rate is defined by the inflow, therefore to represent the infiltration we used these high values of vertical conductivity for the reactive barrier. In fact, since no head was measured at the reactive barrier, results are virtually insensitive to K.

The porosity of the material is applied to the elements of the layers. The 1D elements only represent vertical hydraulic connection between layers, but they overlap with the layers in our model.

Where layer 9 is not present (i.e. everywhere but at the recharge basin) the areal recharge, if any, goes directly to layer 7 in the local scale domain or to layer 1 in the large scale domain.

The code minimizes an objective function that considers the differences between measured and calculated heads, concentrations, and model parameters. For the latter "measured" should be interpreted as prior estimate, in a Bayesian sense. Therefore, RMSWE of model parameters can be interpreted as a measure of plausibility (Medina and Carrera, 1996).

A line including the input mass will be added at table 1 in the revised version, the values are 6683g, 6853g, and 7885g respectively.

**R2.37** Figure 3: - Why does the tracer arrive earlier at P10 than P9, although the first lies behind the latter? I would expect the opposite. (You mention that in P19, L26ff, but do not explain the reason or state a possible explanation.) - It is hard to compare the breakthrough curves, as I could not find information on the filtered area of P2-P10. Please add this information. - Can you explain the unusual tailing of the measurements in P5? - Did you consider effects from double porosity to explain the long tailing of P8.1 and P9?

**R2.37**- From the breakthrough curves it could be assumed that monitoring the point P10 is better connected to the infiltration basin area than the monitoring point P9. Regarding piezometers, see R2.10. Double porosity is an effective model representation of the effect of heterogeneity that was not explicitly included in the model. We acknowledge that horizontal heterogeneity was probably present in the system but not included in the model. We conclude that although its effects could have been included using MRMT (a sophisticated version of double porosity) model, we felt it would lead to a less robust model. Long tails are a consequence of (1) heterogeneity, (2) the complex flow field, and (3) time variability. We try to convey this in Fig. 4.

**R2.38** Figure 4: - The grey areas, should most likely show concentrations of ~0 mg/l, but the legend does not show a grey value. I guess that this is due to lighting settings in Paraview (looks as if you were using this software to produce the figures). Can you please check this?

**R2.38**- Yes, the white area was colored to grey to make it easier to identify the different layers, but corresponds to value 0mg/l.

**R2.39** Figure 5: - From which year is the data you show here? - The legend says "Homo", in the text you defined "Hom" to be the homogeneous case.

**R2.39**-Indeed, the legend has been changed to Hom and we have added the year (2011) in the revised manuscript.

**R2.40** P20, L4: "...because the aquifer transmissivity in the local domain was ultimately the same." - I do not understand this; I thought, that your model had the same setup (i.e. layers have same size); now, if you use different hydr. conductivities, you should end up with different transmissivities. Can you please explain this (I am probably misunderstanding this)?

**R2.40**- We meant that the total transmissivity of the seven layer resulted in a similar values in the three models.

**R2.41** Questions on the model: - Do you define layer thickness through transmissivity of the layers? - What happens, when a solute in a top layer reaches the boundary of the local domain? Is it moving through the outermost vertical elements to the lower layer? Does this vertical travel increase spreading of the plume? - Can you please provide a full list of parameters (eg dispersivity, diffusion coefficients...)?

**R2.41**-We defined layer thickness to accomplish a vertical discretization but it was not based on the transmissivity. The layers are linked with 1D elements that in the edges of the local domain were conferred with a high transmissivity to merge the local domain with the large scale domain avoiding the barrier effect. Therefore, when the solute of one layer reaches the boundary local domain the concentration is transferred to the layer-1 that covers the whole large domain. Dispersivity and diffusion coefficients were not calibrated and they were kept equal for the three models. Longitudinal dispersivity was set to 50 m in the large scale domain, 5m in the 2m-thick layers of the local scale domain, and to 0.3 m in the two 0.3m-layers of the local scale domain. Transverse dispersivity was set to 1.3 m in every layer. Molecular diffusion coefficient was set to $10^{-10} m^2/d$ in the whole domain with the exception of 1D-elements representing the wells (10 $m^2/d$) and the 1D-elements of the local domain edge (10000 $m^2/d$).

**R2.42** General questions: - What is the distance to the groundwater table? I am asking to understand whether it would be important to include the passage through the unsaturated zone. - What was your calibration strategy? Did you conduct a manual or automatic calibration? Which parameters did you use for calibration (TCE, EC, amino-G, heads) and what was the target function (RMSE)? This is important to know in order to evaluate, for example, whether there could actually be more than two fitting calibrations (currently het-1 & het-2), and whether the current conceptual setup may be the reason for non-uniqueness. - The heterogeneous setup clearly shows a better performance (at least for the amino-G tracer). To what extent, do you think, should this heterogeneity be reproduced in the model? Would you think that 3 or 300 layers showed similar results? How could one determine the necessary degree of heterogeneity (or the "sensitivity" of heterogeneity)?

**R2.42**- The groundwater table was between 4 and 6m below the infiltration basin surface. The calibration was automatic (see R2.36). TCA and EC are simulations, non parameter was calibrated, we just used them to validate the calibrated flow and transport parameters for the local scale domain.

**R3.1** The authors investigated a MAR system near Barcelona, Spain using tracer methods combined with numerical analyses. Here they highlighted the importance of heterogeneous structures on the estimation of travel times of waters below an infiltration basin which is of importance when it comes to assessment of removal rates of unwanted substances. The manuscript is clearly written and introduction fits to the content. The methods used and the results obtained are of high scientific relevance. Beside this, the methods and analyses used seem appropriate regarding the objectives focused on. Therefore, I suggest to accept the manuscript after some minor revisions by authors.

**R3.1**- We thank the reviewer for his/her constructive comments and the help improving the manuscript.

Please see the following comments:

**R3.2** The main drawback of the manuscript in its current state is the simplified explanation of the model set-up used here. Although the methods seem scientifically correct, additional information should be provided to the reader to support a better understanding of the content and to ensure reproducibility of the numerical results. This holds true for both the input parameters used here and the numerical procedure. Regarding the first, for example, dispersivities in horizontal and transverse directions cannot be found in the manuscript. However, often dispersivities are used to include the effect of mixing and spreading if spatial variability of the hydraulic parameters is not included directly in the model. This may lead to larger dispersivity values used in homogenous models as compared to heterogeneous models for the same conditions. Are dispersivities for both models the same?

Regarding the numerical procedure some additional information would be helpful, e.g. the software used and how the inner region was selected in its extents (why basin so close to local domain boundary). Beside this, for me it is not clear if and in case of any which retention model was used for the unsaturated zone modeling?

**R3.2** The spirit of this comment is similar to that of the reviewer 2. We must say from the outset that, in the current debate about reproducibility, we do not believe a full description of the model is possible. For example, the overall direction of the plume depends on boundary fluxes and heads, and areal recharge. All of them are spatially and temporally variable. We outline how we obtained them, but not provide the details. These details would be too long to describe. This is especially true for boundary fluxes and recharge, which involve hydrometeorological mass balance (including sources of meteorological stations, soil parameters, etc.) . Providing the time functions describing that variability (some 10 functions) would also be long. Moreover, describing all that information would lead to a boring and tedious paper (it might read more like a report), which would contribute to hide the main message.

We acknowledge the reviewers right (and obligation) to question and we hope that, at least, the local scale domain is fully described. Still, it would be difficult to reproduce. Additional details on the model are provided in the responses to review 2. For some of them, we will cite the discussion in HESSD, rather than lengthening the paper.

Dispersivities have been kept equal for the three models and were established taking into account the dimension of the elements. Longitudinal dispersivities were set to 50 m in the large scale domain, 5m in the 2m-thick layers of the local scale domain, and to 0.3 m in the two 0.3m-layers of the local scale domain. Transverse dispersivities were set to 1.3 m. We must add that dispersivities were not critical in this model (we never felt the need to modify the initial value, which may explain why we forgot to report them). We argue in the paper that, besides heterogeneity (i.e. layering in our model), complex flow geometry (see Figure 4) and time fluctuations contribute to spreading and mixing.

The code used for calibration is TRANSDENS (Hidalgo et al., 2004), which is a development over TRANSIN (Medina and Carrera, 1996; Medina and Carrera, 2003). They employ the Finite Element Methods to solve the equations governing coupled flow and transport in porous media.

The model geometry was based in a regional model of the Llobregat delta main aquifer (Abarca et al.,2006). The inner region, the triangular shape where we defined the multilayer domain, is a particular transmissivity zone defined in the regional model. To reproduce the residence time in the unsaturated zone we used both the reactive barrier porosity and, in part, layer-7.

We have added information regarding the code and the selection of the local scale domain in the revised manuscript.

**R3.3** Next to the mentioned issues, the link between lithology data from the site and the layering used in the models should be better explained (maybe using some lithology profiles as shown in Fig. 1).

**R3.3** The 14m-thick aquifer was divided in seven 2m-thick layers in the local domain aiming to emulate the material differences of the alluvial deposits. The number of layers and their thickness was defined to obtain a sufficient precision in the vertical discretization, but not based in the real lithology which is not expected to be uniform and continuous in the whole domain. Two other additional 0.3 -thick layers with the infiltration basin geometry were added in the local domain to represent the reactive barrier.
The gamma-ray profile was used just to identify a clay layer presents in some monitoring points. Therefore, there is no direct link between the layering and the actual lithology (with the exception of the two 0.3 m-thick layers emulating the reactive barrier).

**R3.4** In page 6, Line 12 and page 8 line 9-10 preferential flow paths were mentioned which indicate non-continuous layering. Are there additional bore profiles, than shown in Fig. 1 supporting the assumption that layers are consistent at the site? Is there any change of thickness in the layers with space in the model? Please explain this further as the fact that every layer is exact 2 m in thickness everywhere sounds a little bit subjective.

**R3.4** The model layers are indeed 2 m-thick along the whole local domain as we mentioned in R3.3. They were defined to emulate the natural layering but not to reproduce the exact geology of the site. From general knowledge of this aquifer, and alluvial aquifers in general, we are convinced that layers are present and discontinuous. The preferential flow path was suggested by the redox indicators species measured in the infiltration basin, the suction cups, and monitoring point P8.3 and discussed in Valhondo et al., 2014, and 2015. We considered the preferential flow is caused by two reason. The heterogeneity of the reactive barrier and the different permeability between the reactive barrier and the sandy medium which favor the infiltration water to flow along fingers that travel a velocity equal to the hydraulic conductivity (Hill and Parlange, 1972; Selker et al., 1996; Cueto-Felgueroso and Juanes., 2008). To emulate these preferential flow paths we distributed the infiltration water volume entering into the infiltration basin, which varied in time and therefore we described it as a time function, between the layer-9 (representing the reactive barrier) and the surface area of the infiltration basin in the layer-7 (representing the medium).

Specific comments:

**R3.5** Page 1, Line 5: "broad" seems rather undefined. Please better define.

**R3.5** We actually mean broad residence time distribution. Zimmerer, C.C., and Kittke, V., 1996 described a wide residence time distribution using the same adjective. We prefer broad over wide or extensive as we are referring to the range of curves shown in Figure 3. But, generally speaking, by broad we mean with a tail much larger that peak arrival time.

**R3.6** Page 1, Lines 7-9: Please check the order of these two sentences. The heterogeneous model is mentioned after the 9 layers which are part of this model.

**R3.6** The general model comprises nine layers, in the homogeneous one all seven layers emulating the aquifer have the same hydraulic properties whereas in the two heterogeneous models these properties varied from layer to layer. In the next sentence "Two type of hypotheses were considered: homogeneous (all flow and transport parameters identical for every layer..." we mentioned the layers, therefore we think that the fact that the model consists of several layers must be addressed before.

**R3.7** Page 2, Line 8: "the wells" – At this point nobody knows about the observation wells of this study.

**R3.7** We are describing the general process, we did not mean the monitoring points of our site but general wells. We write "pumping wells" instead of "the wells".

**R3.8** Page 2 Line 9-10: What is the difference between point (1) and (2): In both cases the concentration of substances are observed and conclusions are made.

**R3.8** True, but in point (1) we refer to the direct observation of removal (which indeed allows deriving some conclusions) and in point (2) we refer to interpretation through transport and kinetic rate models (which also yields concentrations, but of a different nature). We modified the statement to be more explicit about the distinction.

**R3.8** Page 2, Line 14: Flux distribution may also be influenced by temporal flow field changes and different sources and sinks of water balance within a region.

**R3.9** Indeed, we include this in the "complexity of natural systems".

**R3.10** Page 2, Line 17-19: Vertical distribution of hydraulic properties can be gained among others using Direct-Push techniques (e.g. Dietrich et al. 2008, Butler et al. 2002).

**R3.10** Indeed, but even with the Direct-Push technique it is unfeasible to characterize the small scale variations of hydraulic properties. We have added a sentence regarding this in the introduction.

**R3.11** Figure 1: The gamma-ray profile and the lithology should be shown in Fig 1B.

**R3.11** Fig. 1B shows horizontal location and basins and monitoring points situation, and Fig 1C is related to the geology, therefore we think the gamma-ray profile and the lithology matched here.

**R3.12** Page 5, Line 22: Is the large domain the regional model sometimes mentioned? Please clarify procedure here.

**R3.12** The large scale domain is the one delimited by white lines in Figure 1A, This large scale domain is a portion of the regional model of the Llobregat delta main aquifer (Abarca et al.,2006). The large scale domain is limited by the geological natural edges of the fluvial valley (that are the limit of the regional model too) and by two batteries of piezometers perpendicular to the regional flow used to prescribe a Dirichlet type boundary condition. This large scale domain consist in one single layer and maintained the different transmissivity, land uses, recharge areas...defined in the regional model (Abarca et al., 2006). To increase the detail around the artificial recharge system and include the vertical variations we divided the thickness of the aquifer in a particular transmissivity zone defined in the regional model in

seven 2m-thick layers, and named this multilayer portion "local domain". The large scale domain and the local scale domain are therefore totally coupled, both are solved together in every model run. The division is made only for practical reason. We have modified the methods sections of the revised manuscript willing to clarify the procedure.

**R3.13** Page 5, Line 30: Which kind of local tests do you mean?

**R3.13** A pumping test performed when a well located close to P5 was drilled and a tracer test with radial flow performed before the installation of the reactive barrier, both will be mentioned in the revised manuscript.

**R3.14** Page 6, Line 10: How this function look like? Is there an areal distribution of the input concentration?

**R3.14** The time function was built to introduce to total amount of amino-G acid mass in 15 min. The infiltration basin was divided in nine zones and for each of them a factor that multiplies the time function was calculated. This factor was higher than 1 in the zones closer to the pipe that pours water to the infiltration basin and less than 1 in the rest zones. Mass balance was taken into account in these factors calculation to ensure that the total amino-G mass introduced was precise. The introduced mass for each outcome (Het-1, Het-2, and Hom) will be included in Table 1 in the revised manuscript.

**R3.15** Page 6, Line 14: Why both layers, layer 7 and 9 were used distributing the timedependent inflow data?

**R3.15** To emulate and estimate the preferential flow through the reactive barrier (see response R2.22 and R2.26).

**R3.16** Fig. 3: Different scales make the concentrations hard to compare with each other. Maybe these could be transformed into equal scales (maybe using logarithmic scales?)?

**R3.16** The reviewer is right and we devoted some time to this issue. We tried logarithmic scales but the main objective was to compare the performance of the three outcomes between them and their fit with the observation in each monitoring point. However, given the range of concentrations, we had to use three cycles (if we wanted the same range for all wells). Using the same arithmetic scale, it was hard to see anything in the low concentration wells. So, in the end, we decided to use arithmetic scale adjusted differently for each well. The time scales are different as well, so we decided to use regular scale for time and concentrations.

**R3.17** Page 11, Line 7: typo "iwas"

**R3.17** It has been corrected in the revised manuscript, thank you.

**R3.18** Page 11, Line 25 Is TCA possibly subject to reaction (conservative modeling used here)?

**R3.18** Citing Lookman et al., 2004 " Through abiotic reduction by common hydrolysis in water TCA is transformed to DCE and acetate (following first-order kinetics with half-life of 2.9 years at 15ºC)". Together with the TCA, DCE was measured in all collected samples and DCE concentration was always bellow detection limit (<8 µg/l). The dissolved organic carbon

concentration in the native groundwater averaged 2.5 mg/l, and it did not display variations between the different monitoring points indicating that probably the bioavailable dissolved organic carbon was negligible and little biodegradation might be expected. This together with the fact that the residence time in the local scale domain is less than six months (lower than the half-life time proposed for the TCA), we treated the TCA as a conservative tracer.

**R3.19** Page 11, Line 33-34: Maybe velocity field without artificial recharge is not very sensitive to local hydraulic conductivities, but maybe other processes were missed in the model as it was built and calibrated to reproduce the vertical infiltration processes.

**R3.19** We agree. It is highly likely that there are processes that we are not taking into account.

**R3.20** Fig. 5 and 6 and statements on page 11 line 32ff: What is the screened portion of the Px named observation wells? Does the screened interval match with the lithologic layering used in the model (for all observation points, also P8.x wells) and the respective concentrations simulated in the layers?

**R3.20** Monitoring points P8.1, P8.2, and P8.3 were screened at 13-15m, 10-12m, and 7-9m below the infiltration basin surface. The other four monitoring points were fully screened. The layer with high transmissivity and highest head is the one that controls observed concentrations. Since recharge generates downloads flux, we expected concentrations to be controlled by the highest T layer.
The layers defined on the model were aimed to emulate the heterogeneity of an alluvial aquifer but they are not a exact representation of the local geology. Therefore, we did not know a priori which layer controlled observed concentrations and we had to perform a screening with different combinations of layers for the four totally screened piezometers and obtain the better match for each one, which are the layers displayed in Figure 4. This procedure has been included in the revised manuscript.

**References**

- Abarca, E., Vázquez-Suñé, E., Carrera, J., Capino, B., Gámez, D., and Batlle, F.: Optimal design of measures to correct seawater intrusion, Water Resources Research, 42, n/a–n/a, doi:10.1029/2005WR004524, w09415, 2006.

- Carle, S. F. and Fogg, G. E: Modeling Spatial Variability with One and Multidimensional Continuous-Lag Markov Chains, Mathematical Geology, 29, 891-918, 1997. doi: 10.1023/A:1022303706942.

-Cueto-Felgueroso, L., Juanes, R.: Nonlocal interface dynamics and pattern formation in gravity-driven unsaturated flow through porous media. Phys. Rev. Lett. 101, 244504. doi:10.1103/PhysRevLett.101.244504, 2008

- Gámez, D., Simó, J. A., Lobo, F. J., Barnolas, A., Carrera, J., & Vázquez-Suñé, E. (2009). Onshore–offshore correlation of the Llobregat deltaic system, Spain: Development of deltaic geometries under different relative sea-level and growth fault influences. Sedimentary Geology, 217(1), 65-84.

- Hidalgo, J. J., Slooten, L., Medina, A., and Carrera, J.: A Newton-Raphson based code for seawater intrusion modelling and parameter estimation, Groundwater And Saline Intrusion Selected Papers from the 18th Salt Water Intrusion Meeting, 18th SWIM, edited by Hidrogeologia y Aguas Subterraneas IGME, Madrid, S., 15, pp. 111–120, IGME, doi:84-7840-588-7, 2004.

-Hill, D.E., Parlange, J.Y., 1972. Wetting front instability in layered soils. Soil Sci. Soc. Am. J. 36, 697–702, doi: 10.2136/sssaj1972.03615995003600050010x, 1972

- Lookman, R., Bastiaens, L., Borremans, B., Maesen, M., Gemoets, J., and Diels, L.: Batch-test study on the dechlorination of 1,1,1-trichloroethane in contaminated aquifer material by zero-valent iron, Journal of Contaminant Hydrology, 133–144, doi: 10.1016/j.jconhyd.2004.02.007, 2004.

- Medina, A. and Carrera, J.: Coupled estimation of flow and solute transport parameters, Water Resources Research, 32, 3063–3076, doi:10.1029/96WR00754, 1996.

- Medina, A. and Carrera, J.: Geostatistical inversion of coupled problems: dealing with computational burden and different types of data, Journal of Hydrology, 281, 251 – 264, doi:10.1016/S0022-1694(03)00190-2, 2003.

- Park, H., Cha, D., Fox, P.: Uncertainty Analysis of Mound Monitoring for Recharged Water from Surface Spreading Basins, Journal of Environmental Engineering, 132, 1527-1579, 2006. doi: 10.1061/(ASCE)0733-9372(2006)132:12(1572).

-Selker, J.S., Steenhuis, T.S., Parlange, J.-Y.: An engineering approach to fingered vadose pollutant transport. Geoderma 70 (2–4), 197–206, doi: 10.1016/0016-7061(95)00085-2, 1996.

- Thomson, A. F. B., Carle, S. F., Rosenberg, N. D., Maxwell, R. M.: Analysis of groundwater migration from artificial recharge in a large urban aquifer: A simulation perspective. Water Resources Research, 35, 2981-2998, 1999. doi: 10.1029/1999WR900175

- Valhondo, C., Carrera, J., Ayora, C., Barbieri, M., Nödler, K., Licha, T., and Huerta, M.: Behavior of nine selected emerging trace organic contaminants in an artificial recharge system supplemented with a reactive barrier, Environmental Science and Pollution Research, 1–12, doi:10.1007/s11356-014-2834-7, 2014.

- Valhondo, C., Carrera, J., Ayora, C., Tubau, I., Martinez-Landa, L., Nödler, K., and Licha, T.: Characterizing redox conditions and monitoring attenuation of selected pharmaceuticals during artificial recharge through a reactive layer, Science of The Total Environment, 512–513, 10 240 – 250, doi:10.1016/j.scitotenv.2015.01.030, 2015.

- Valhondo et al.: Reply to comments by Prof. Marc Walther, Interactive discussion HESSD, Url: http://editor.copernicus.org/index.php/hess-2016-197-AC2.pdf?_mdl=msover_md&_jrl=13&_lcm=oc108lcm109w&_acm=get_comm_file&_ms=51089&c=108879&salt=1312152279648298862, 2016

- Walther, M.: Comments on "Tracer test modeling for local scale residence time distribution characterization in an artificial recharge site", Interactive discussion HESSD, Url: http://editor.copernicus.org/index.php/hess-2016-197-

RC2.pdf?_mdl=msover_md&_jrl=13&_lcm=oc108lcm109w&_acm=get_comm_file&_ms=51089&c=108491&salt=1257066553820224553, 2016.

- Zimmerer, C. C., and Kottke, V.: Effects of spacer geometry on preassure drop, mass transfer, mixing behavior , and residence time distribution, Desalination, 129–134, doi:10.1016/0011-9164(96)00035-5, 1996.

- Zhao, S., Chang, D., Jianzhong, H.: Detoxification of 1,1,2-Trichloroethane to Ethene by Desulfitobacterium and Identification of Its Functional Reductase Gene, PLoS ONE, 1-13, doi: 10.1371/journal.pone.0119507, 2015.

---

## Author Response (AR2)

Responses to comments by Prof. Marc Walther.

Dear Prof. Marc Walther, we thank you for the time devoted to this manuscript that has lead to a significant improvement of the manuscript. We have implemented some of the suggested information and changes to the new document.

R2.5 and fig. 1: Thank you for clarification.
- Concerning the blue small arrows for text reference: please use them consistently, either pointing from the text to the object or vice versa (compare arrows for "Recharge Basin" and "Quaternary Terrace"); also, not all arrows are clear: on the right, does the arrow show the location of "Aluvial Fan" or the direction of "Lateral Inflow"?
- Concerning the stratigraphical log: I understand your intention to add some more, but not too much, background information for the reader. I think, however, that with the current state of the log figure, it is too less explanation to be valuable: I see a legend for a figure as compulsory, and additionally, the figure is in very low resolution so I cannot identify the texture to guess the character of the layers. I really think it would be better to add more details here. Maybe, you can add text to the layers (as you did with "Clay" at 17m)?

R2.5 You are welcome. The small arrows of Fig 1 has been changed to be consistent. The stratigraphical log has been removed as the arrows showing the lateral inflow directions.

R2.26 Concerning your answer "We used abbreviation "Ly" because it is the same nomenclature displayed in Figure 4." - I could not find an explaination of this abbreviation (although it is clear to me, it may not be for all readers). Maybe, you can simply add this to the caption of figure 4?

R2.26 We have been added the abbreviation in the caption of the figure.

R2.34 I still cannot find information about which model setup you show in figure 2 (homogeneous, het1 or het2)? Can you please add this?

R2.34 The model of Figure 2 corresponds with the large scale flow model, the fit are those obtained after the recalibration of the flow parameters. Afterwards we estimated the transport parameters under the two hypotheses, so the model of figure 2 is the base for the Hom, Het-1 and Het-2.

R2.35 You write "horizontal and vertical diffusion coefficients" - do you mean dispersion coefficients? I can find information on the dispersivity (P6L6).
Can you please add information on the diffusion coefficient?

R2.35 Yes, I mean dispersion coefficients, sorry. The diffusion coefficients remains equal for the three models, $10^{-9}m^2/d$ for the whole domain with the exception of the 1D elements linking the local and the large scale domains for which the diffusion is $10^3 m^2/d$ and the 1D elements representing wells where it was fir to $10\,m^2/d$. This information has been added to the manuscript.

R2.36 Thank you for the detailed answer! I suggest that you add your description of the RMSWE value to the manuscript, as it is relatively uncommon how you calculate this (also considering the high value of it). Is there a unit for RMSWE or is it unitless through

normalization of the parameters/values? Frankly speaking, an equation or reference would be nice for the RMSWE (to better understand and interpret the value you show in table 1).

R2.36 Yes it is dimensionless, in the cited bibliography it is possible to find detailed description of the RMSWE.

R2.38 Your answer "Yes, the white area was colored to grey to make it easier to identify the different layers, but corresponds to value 0mg/l." - so why don't you change the legend to show a grey value for 0mg/l?

R2.38 We have changed the legend in Figure 4.

R2.40 The phrasing of "because the aquifer transmissivity in the local domain was ultimately the same." is still used. I understand your answer in the reply, but am in doubt that readers may misinterpret this in the first place.

R2.40 We have changed the sentence as it appeared in the response to comments.

R2.41 Is it right, that you set a molecular diffusion coefficient of 10e-10, 10, and 10000 m2/d, for the domain, 1D near the wells, and at the edge of the local domain??? That would be about 1e-15, 1e-4, and 1e-1 m2/s, which is in any case far from what is reported to be a common value for diffusion of a solute in water ~2e-9 m2/s (not even speaking of an effective diffusion coefficient in porous media (but this is probably not important))! Please, explain your choice of diffusion values here.

R2.41 It is right they are a very small values , around 10e-13 m2/s.  The actual value of the diffusion coefficient is only important for the wells and the 1D elements connecting the layers because where the mixing of solutes in the thickness of the layer (or along the well) has to be accounted for. In the rest of the domain the model is not sensitive to diffusion because dispersion is the dominant mechanism.

P7 L29: "mean weighted residual for head observations" should have a unit (meter?).

P7 L29 It is dimensionless.

P15 L24: "ACWAPUR (XXXXX)" - is there a number/reference missing?

P15L24  The reference was missing, it has been added in the new manuscript.

Responses to comments by reviewer 2.

the new manuscript looks very good and I still want to highlight the importance of the work done here.

The authors thank reviewer 2 for the help improving the manuscript.

I only have some very minor comments the author may check before publication:

page 2, line 24: Maybe "feasible" is not the rigth wording, please add "often" or change wording to "cumbersome".

It has been changed

page 4, line: 10: "cause" instead of "causes"

Done

page 7, line 12: "risk 'of' overparameterization"

We have corrected it

page 7, line 29: Are there units missing?

No, RMSWE is dimensionless

Figure 2: What are the numbers in the right hand side? Did I miss something?

I apologize, I attached the wrong figure, it has been corrected in the new manuscript.

Figure 3 caption: "monitoring 'of' the amino-G acid"

Done

page 12, line 33: "did" instead of "dis"

It has been corrected

page 15, line 16: What is "much"?

Efficiency

page 15, line 18: please change "direct-push" to "direct-push based exploration"

It has been changed